# Finite-size local dimension as a tool for extracting geometrical properties of attractors of dynamical systems

Martin Bonte[1] and Stéphane Vannitsem[1]

[1]Royal Meteorological Institute of Belgium, Brussels, Belgium

**Correspondence:** Martin Bonte (martin.bonte@meteo.be)

**Abstract.** Local dimension computed using Extreme Value Theory (EVT) is usually used as a tool to infer dynamical properties of a given state $\zeta$ of the chaotic attractor of the system. The dimension computed in this way is also known as the pointwise dimension in dynamical systems literature, and is defined using a limit for an infinitely small neighborhood in the phase space around $\zeta$. Since it is numerically impossible to achieve such a limit, and because dynamical systems theory predicts that this local dimension is almost constant over the attractor, understanding the properties of this tool for a finite scale $R$ is crucial. We show that the dimension can considerably depend on $R$, and this view differs from the usual one in geophysics literature, where it is often considered that there is one dimension for a given dynamical state or process. We also systematically assess the reliability of the computed dimension given the number of points to compute it.

This interpretation of the $R$-dependence of the local dimension is illustrated on the Lorenz 63 system for $\rho = 28$, but also in the intermittent case $\rho = 166.5$. The latter case shows how the dimension can be used to infer some geometrical properties of the attractor in phase space. The Lorenz 96 system with $n = 50$ dimensions is also used as a higher dimension example. A dataset of radar images of precipitation (the RADCLIM dataset) is finally considered, with the goal of relating the computed dimension to the (in)stability of a given rain field.

## 1 Introduction

When nowcasting the rain field, the future state is essentially predicted using Lagrangian persistence (Zawadzki et al. (1994), see Pierce et al. (2012) for a review on nowcasting). It is known that the errors in the estimation of the motion field (the wind) do not dominate the total error of the forecast (Bowler et al., 2007), but that taking into account the growth and dissipation of rain cells is essential for an accurate nowcast (Germann et al., 2006). In the case of convective events, the instability may be captured by the convective available potential energy (CAPE) available in NWP outputs, but convective situations can very quickly evolve. It would therefore be useful to have a real-time method to assess the stability of the current situation. Several techniques have been developed to produce probabilistic forecasts (Germann and Zawadzki, 2004; Bowler et al., 2007; Berenguer et al., 2011; Pulkkinen et al., 2019a). Despite these, it is still difficult to emit early warnings for very severe floods for example, as witnessed by the 2021 flood in Belgium (Journée et al., 2023).

The idea behind the current work is to use the local dimension of a given state in phase space, as a proxy for the complexity (and possibly the predictability) of the future of that state, following Faranda et al. (2017, 2022, 2023); De Luca et al. (2020).

The intuition supporting this idea is that points having a high dimension have a lot of different phase space directions in which to evolve on the attractor, so that their direction of evolution would be more difficult to guess if one had to do it stochastically. Other phase space ideas for nowcasting have also been explored in Foresti et al. (2024).

The computation of the dimension of manifolds and attractors of dynamical systems is a broad and old topic (Russell et al. (1980), see Abarbanel (1996); Ott (2002) for textbook reviews in the context of dynamical systems, and Camastra and Staiano (2016) for a review on the dimension in the broader context of manifold dimension estimation). New algorithms were proposed for low dimensional systems, producing mainly global estimation of the dimension (Golay and Kanevski, 2014; Erba et al., 2019; Bac and Zinovyev, 2020). The focus is here on the dimension computed locally in phase space, using the framework of the extreme value theory (EVT), as proposed in Faranda et al. (2012, 2017, 2019, 2023); Pons et al. (2020, 2023).

The local dimension is also called the pointwise dimension, and its definition is (Ott, 2002)

$$D_p(\zeta) = \lim_{R \to 0} \frac{\ln \mu(B(R))}{\ln R}, \tag{1}$$

where $\mu$ is a given measure on the attractor and $B(R)$ is the $n$-dimensional ball centered on $\zeta$. This definition implies that $\mu(B(R)) \overset{R \to 0}{\sim} R^{D_p(\zeta)}$ asymptotically (limits of both infinite number of points and infinitely small bulk size). A classical result in dynamical systems theory is that, if $\mu$ is ergodic, the dimension is asymptotically the same for all points, $D_p(\zeta) = D_1$ ($D_1$ is the information dimension), except for a zero-measure set of points (Pons et al., 2020; Ott, 2002; Pesin, 1997).

It is obviously impossible to numerically reach the limit $R \to 0$ in (1), and the question is then how to choose some finite value of $R$, as raised for example in Pons et al. (2020). Datseris et al. (2023) suggests that $N$ (the number of points at a distance smaller than $R$ from the computation point $\zeta$) just needs to be higher than 100-1000. A similar conclusion is reached in Caby (2019). The idea of evaluating the impact of finite radius $R$ was followed in Little et al. (2017) using a PCA technique in order to identify a scale where the manifold could be approximated by a plane. A similar idea was used to evaluate the robustness to multiscaling of a good estimator in Camastra and Staiano (2016).

In this work, we use a maximum likelihood estimator to estimate the local EVT dimension, for different values of $R$. This estimator turns out to be exactly a local version of the Takens estimator for the global dimension (Takens, 1985). The estimations of the dimension for the different values of $R$ are then used to infer local information on the attractor.

As the focus of the current work is on the non-asymptotic estimation of the dimension, we obviously want to consider only values of the dimension which will not significantly change if more points are added to the dataset: in this sense, we want to be in the limit of infinite number of points. We therefore need some techniques to assess whether the dimension computed for some value of $R$ has sufficiently converged or not.

The main findings of this work are:

1. The non-asymptotic local dimension depends on the scale $R$: this can be used to get information on the phase space structures. This new interpretation of the dimension as dependent on $R$ contrasts with the usual notion of dimension. We illustrate this approach on the Lorenz 63 system for $\rho = 28$ and for $\rho = 166.5$. The latter displays chaotic intermittency (see Sparrow (1982); Ott (2002)), inducing a phase space geometry well suited to illustrate our interpretation of the

dependence of the dimension on $R$. Another reason to study the behavior of the dimension for intermittent systems, comes from the fact that our main goal is to compute the dimension for rain fields, which are known to be intermittent.

2. The question of the number of points $N$ needed to have a robust estimation of the dimension is also explored. The question is raised in Pons et al. (2020) and Datseris et al. (2023) for the EVT dimension. For the correlation dimension (Grassberger and Procaccia, 1983), the definition of the dimension ($size \sim R^{dimension}$) implies, at fixed $R$, that the needed value of $N$ grows exponentially with the dimension (Eckmann and Ruelle, 1992; Camastra and Staiano, 2016).

The normalized root mean squared error (NRMSE) (metric proposed in Datseris et al. (2023)) is used in this work to assess the quality of the fit at a given scale $R$. The confidence bounds provided by the likelihood function are also computed and it is shown that the true value of the dimension is around the estimated value with a $10\%$ error and with $95.5\%$ confidence if $N > 427$, seemingly in contradiction with the mentioned argument of Eckmann and Ruelle (1992). We introduce a quantity (denoted $s$ hereafter) which brings some light on this. This gives a possible answer to the question of the maximal dimension that one can measure with the EVT method and under which conditions.

3. The applicability to high-dimensional systems is also explored in the context of the Lorenz 96 with $n = 50$ dimensions, for which the dependence of the dimension on $R$ gives some characterization of the phase space.

Radar-estimated rain field over Belgium is investigated with the same tools. In this case, the dimensions for some computation points $\zeta$ can be reliably estimated, but only for very narrow ranges of $R$, and this makes it difficult to draw conclusions on the dynamical properties of the state based on the values of the dimension. The dimension ranges essentially between 10 and 30, with a lot of values between 15 and 20. Not surprisingly, we also find a correlation between the convective rain rate from ERA5 reanalysis and the relevant range of values of $R$.

This paper is organized as follows. In section 2, we first introduce the pointwise dimension and show how it is computed in the EVT framework. We also derive the expression of the maximum likelihood estimator of the dimension and of the bounds of the confidence interval using the likelihood function. After that, we introduce the NRMSE score and interpret the estimated dimension. We also introduce the quantity $s$ in order to understand the maximum dimension that one can compute using a given number of points.

Section 3 introduces the results of the computed dimension on the Lorenz 63 system (for $\rho = 28$ and $\rho = 166.5$) and discuss in details the interpretation. The intermittent case illustrates well that the dimension depends on the radius $R$, and can be used to evaluate the geometric properties of the attractor, such as the distance of chaotic points to the laminar regime.

Large systems (Lorenz 96 with $n = 50$ dimensions and the radar dataset) are considered in section 4 and the last section briefly summarizes the work.

## 2 Framework for the estimation of the dimension

Given a measure on the attractor, the pointwise dimension for a computation point $\zeta$ is defined in terms of the natural measure $\mu$ on the attractor by (1):

$$D_p(\zeta) = \lim_{R \to 0} \frac{\ln \mu(B(R))}{\ln R}.$$
(2)

This is equivalent to

$$\mu(B(R)) \overset{R \to 0}{\sim} R^{D_p(\zeta)}$$
(3)

i.e. the measure of the ball $B(R)$ of radius $R$ around $\zeta$ scales as $R^{D_p(\zeta)}$.

The measure $\mu$ is often chosen to be the natural measure of the system: given any long enough trajectory originating from a typical initial condition of the system, $\mu(A)$ is defined for any subset $A$ of the phase space as the fraction of points of the trajectory inside $A$, or equivalently as the fraction of time spend by the system in $A$ (see Ott (2002); Kantz and Schreiber (2003)). This means that the whole set of points of the trajectory is as if it had been sampled from the measure $\mu$. Some points may have to be discarded at the beginning of the trajectory in order to ensure that it is on the attractor.

This natural measure is invariant by definition: $\mu(A) = \mu(\Phi_t^{-1}(A))$, where $\Phi_t$ is the flow of the system. This is because each point in the trajectory has one antecedent point, so that the number of points in A is the same than in $\Phi_t^{-1}(A)$. It is also ergodic if the attractor cannot be decomposed in two distinct invariant sets. In practice, one can also assume ergodicity by assuming that the system is always on the same invariant subset (for example because we observe one and only one trajectory, as in the case of climate). As stated in the introduction, the dimension $D_p(\zeta)$ is constant for almost all points when the measure is ergodic (Pesin, 1997; Ott, 2002; Pons et al., 2020).

The above discussion implies that, given a long enough trajectory, $\mu(B(R))$ can be approximated as the number $C(R)$ of points inside $B(R)$, divided by the total number of points in the trajectory. It follows that the pointwise dimension $D_p(\zeta)$ can be interpreted as a characterization (see (5) below for a precise statement) of the growth rate of the number points $C(R)$ that one should find inside a ball $B(R)$ centered around $\zeta$ (for infinitely small $R$).

If there are enough points around $\zeta$, and if they span a smooth $D_p(\zeta)$-dimensional surface, one can introduce the hyperspherical coordinates $(r, \boldsymbol{\theta})$ and the density of points $\sigma(r, \boldsymbol{\theta})$. In this case, the number of points $C(R)$ inside $B(R)$ is

$$C(R) = \int_0^R dr \, r^{D_p(\zeta)-1} \Sigma(r)$$
(4)

with $\Sigma(r) = \int d\boldsymbol{\theta} \sigma(r, \boldsymbol{\theta})$ being the density integrated over angles. The number of points $C(R)$ is proportional to $R^{D_p(\zeta)}$ if and only if $\Sigma(r)$ is constant. If not, one has to consider the small $R$ limit, where $C$ can be expanded in Taylor series. One can check that the first nonzero term of this series is the one of order $D_p(\zeta)$:

$$C(R) = C^{(\delta)}(0) \frac{R^\delta}{\delta!} + O(R^{\delta+1}).$$
(5)

where the notation $\delta = D_p(\zeta)$ is used from now on. For example, for $\delta = 2$, we have

$$C(0) = 0 \tag{6}$$

$$C'(0) = R\Sigma(R)|_{R=0} = 0 \tag{7}$$

$$C''(0) = (R\Sigma(R))'|_{R=0} = \Sigma(0), \tag{8}$$

so that $C(R) = \frac{R^2}{2}\Sigma(0) + O(R^3)$. This approximation amounts to consider $\Sigma$ constant on the interval of integration $[0, R]$ because the $O(R^3)$-term containing $\Sigma'(0)$ is neglected.

After estimation of $C(R)$ for several values of $R$, and if it has the expected scaling $C(R) \sim R^\delta$, the dimension can be extracted using a fit on this scaling. In the context of the correlation dimension, $C(R)$ is a local version of the correlation integral and $\delta$ can be computed as $\frac{\ln C(R_2) - \ln C(R_1)}{\ln R_2 - \ln R_1}$ ($R_1$ and $R_2$ have to be chosen).

Whatever technique we use to estimate $\delta$, the following points have to be kept in mind:

- The distances measured in phase space do not precisely match with $r$ or $R$: for (5) to hold, $R$ has to be measured along the surface of the attractor (more precisely, along the geodesics of the attractor, provided it can be approximated by a smooth manifold), but this is not possible when our representation of the attractor is a set of points. Instead, we measure distances in phase space and, if the attractor is curved, there can be a mismatch with the distances measured along the surface of the attractor (see Fig. 1). Therefore, the number of points in $B(R)$ might not scale exactly as $R^\delta$. This could lead to a bias in the estimated dimension (Perinelli et al., 2023). Since the distance computed in phase space becomes closer to the one computed on the surface when $R$ is small enough (i.e. smaller than some typical scale of the curvature, given for example by the inverse of the curvature itself), one recovers $C(R) \sim R^\delta$ in the limit $R \to 0$.

- If we use a too big value of $R$, the $R^{\delta+1}$ term in (5) might not be small anymore. Said differently, $\Sigma(r)$ in (4) is not sufficiently constant on $[0, R]$ for $C(R) \sim R^\delta$ to hold. Actually, if in this range, $\Sigma(r)$ looks more like $\sim r^a$ for $a \neq 0$, $C(R)$ will scale as $R^{\delta+a}$. The value of the dimension we will measure in this case does not have a clear geometric meaning, but is an effective value $\delta_{eff} = \delta + a$, taking into account the change of $\Sigma$ over $[0, R]$.

  In this case, the information contained in $\delta_{eff}$ is much more difficult to use. If one had a way to estimate $a$ in $\Sigma(r) \sim r^a$, one could compute $\delta = \delta_{eff} - a$.

## 2.1 Dimension and extreme value theory

Extreme value theory is a framework to study the occurrences of extreme events (see for example the textbooks Beirlant et al. (2004); Falk et al. (2010); Lucarini et al. (2016)).

Two approaches can be followed to define extremes among the samples $\{X_i\}_{i=1,...n}$. The Peak Over Threshold (POT) consists in fixing a threshold $u$, and the values $X_i > u$ above this threshold are considered as extremes. The threshold $u$ has to be taken as high as possible to reach the correct definition of extremes. The Block Maxima (BM) approach consists in splitting the set of samples in chunks of size $m$, and the extremes are the highest values of each chunk (one extreme for each chunk). The size $m$ has to go to infinity in order to correctly define of extremes.

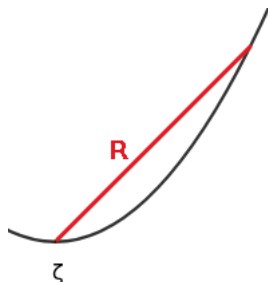

**Figure 1.** Illustration that the distance computed in phase space might be different than the distance on the surface: the distance $R$ computed in phase space is in red, while $R$ should be measured along the surface of which we are trying to estimate the dimension (which is in black), in order to find the $C(R) \sim R^\delta$ behavior.

The main theorem in the BM approach states that, under some conditions, there are essentially three asymptotic distribution for extremes, regrouped under the Generalized Extreme Value (GEV) law. In the POT approach, an equivalent theorem states that there are also three limiting distributions, regrouped under the Generalized Pareto Distribution (GPD):

$$GPD_\xi(z) = \begin{cases} (1+\xi z)^{-1-1/\xi} & \text{for } \xi \neq 0 \\ e^{-z} & \text{for } \xi = 0 \end{cases} \tag{9}$$

where the support of $z$ is $z \geq 0$ for $\xi \geq 0$ and $0 \leq z \leq -1/\xi$ for $\xi < 0$.

### 2.1.1  Theoretical extreme value law around $\zeta$

The scaling $C(R) \sim R^\delta$ induced by the definition of the pointwise dimension (1) is the starting point to estimate the dimension in the EVT framework using the POT approach (Faranda et al., 2012). We reformulate here some of their results.

Consider the ball $B(R)$ of radius $R$ in the $n$-dimensional phase space. $C(R) \sim R^\delta$ implies that the number of points between $r$ and $r + dr$ is

$$c(r)dr \sim C'(r)dr = \frac{C^\delta(0)}{(\delta-1)!}r^{\delta-1}\, dr. \tag{10}$$

In order to have a pdf between $0$ and $R$ (i.e. in the $B(R)$ ball), $c$ is normalized to 1 ($\int_0^R c(r)dr = 1$), so that

$$c(r) = \frac{\delta\, r^{\delta-1}}{R^\delta} \tag{11}$$

With such a normalization for $c(r)$, it can be interpreted as follows: if we select a point randomly inside $B(R)$, $c(r)dr$ is the probability that its distance to $\zeta$ is between $r$ and $r + dr$ (for $r < R$).

The usual EVT framework for the dimension is formulated within the POT approach. The observable whose extreme values distribution is studied is often one of the following functions of $r$:

$$g_1(r) = -\ln r, \quad g_2(r) = r^{-1/\alpha}, \quad g_3(r) = K - r^{1/\gamma}. \tag{12}$$

The parameter $K$ can be freely chosen, and $\alpha$ and $\gamma$ have to be positive. Given a point $\zeta$ in phase space, the points corresponding to extremes are those whose distances to $\zeta$ are smaller than $R$. The threshold of the POT approach is given by $T_a = g_a(R)$ ($a = 1,2,3$), where $g_a$ is the function that was chosen from the above three. Note that EVT usually defines extremes as *high* values of an observable while, in terms of the distance $r$ to $\zeta$, extremes are defined as *small* values of $r$. Using these transformations and their inverses as well as (11), the pdf's describing the distributions of $g_a = g_a(r)$ are computed to be

$$f_{1,T_1}(g_1) = \delta e^{-\delta(g_1 - T_1)}, \quad f_{2,T_2}(g_2) = \frac{\delta\alpha}{T_2}\left(\frac{g_2}{T_2}\right)^{-\delta\alpha - 1}, \quad f_{3,T_3}(g_3) = \frac{\delta\gamma}{K - T_3}\left(\frac{K - g_3}{K - T_3}\right)^{\delta\gamma - 1}. \tag{13}$$

Note that one has always $g_a \geq T_a$ because $r \geq R$ and the functions in (12) are decreasing for $r > 0$. One can check that each of the distributions in (13) correspond exactly to one of the signs of $\xi$ ($\xi = 0$, $\xi > 0$ and $\xi < 0$) of the GPD (9), with

$$
\begin{aligned}
&f_{1,T_1}(g_1)dg_1 = GPD_0(z)dz && \text{with } z = \delta(g_1 - T_1) \\
&f_{2,T_2}(g_2)dg_2 = GPD_\xi(z)dz && \text{with } \xi = \tfrac{1}{\delta\alpha} \text{ and } \tfrac{g_2}{T_2} = 1 + \xi z \\
&f_{3,T_3}(g_3)dg_3 = GPD_\xi(z)dz && \text{with } \xi = -\tfrac{1}{\delta\gamma} \text{ and } \tfrac{K - g_3}{K - T_3} = 1 + \xi z
\end{aligned}
\tag{14}
$$

Note that, starting from one of the three possible distributions in (13) and using the corresponding transformations (12), one recovers back equation (11). That is, the application of each one of the transformations in (12) and the deduction of $\delta$ using a parameter fit of the corresponding distribution in (13), is an alternative way to access the exponent $\delta$ of the scaling $C(R) \sim R^\delta$.

### 2.1.2 Maximum likelihood estimation of $\delta$

The expression of the maximum likelihood (ML) estimator of the dimension is now derived, using the above density functions. It leads in fact to the expression of a local version of the estimator of Takens (Takens, 1985).

To compute the dimension around a computation point $\zeta$, consider all points inside the ball $B(R)$ centered on $\zeta$ ("the analogues"), whose distances to $\zeta$ are $r_i$ (with $i = 1, \ldots, N$ and $r_i < R$). The corresponding $g$'s are defined as $g_{a,i} = g_a(r_i)$. We know from the previous section that the $g_{a,i}$'s should follow the distribution $f_{a,T_a}$.

The (log)-likelihood functions are for each case:

$$L_1 = \prod_{i=1}^{N} \delta e^{-\delta(g_{1,i} - T_1)}, \quad \ln L_1 = N\ln\delta - \delta\sum_{i=1}^{N}(g_{1,i} - T_1) \tag{15}$$

$$L_2 = \prod_{i=1}^{N} \delta\alpha\left(\frac{g_{2,i}}{T_2}\right)^{-\delta\alpha - 1}, \quad \ln L_2 = N\ln(\delta\alpha) - (\delta\alpha + 1)\sum_{i=1}^{N}\ln\left(\frac{g_{2,i}}{T_2}\right) \tag{16}$$

$$L_3 = \prod_{i=1}^{N} \delta\gamma\left(\frac{K - g_{3,i}}{K - T_3}\right)^{\delta\gamma - 1}, \quad \ln L_3 = N\ln(\delta\gamma) + (\delta\gamma - 1)\sum_{i=1}^{N}\ln\left(\frac{K - g_{3,i}}{K - T_3}\right) \tag{17}$$

Setting the derivative of $\ln L$ with respect to $\delta$ to 0 in each case gives the ML estimator:

$$\frac{1}{\hat{\delta}_1} = \frac{\sum_{i=1}^{N}(g_{1,i} - T_1)}{N}, \quad \frac{1}{\hat{\delta}_2} = \frac{\alpha}{N}\sum_{i=1}^{N}\ln\left(\frac{g_{2,i}}{T_2}\right), \quad \frac{1}{\hat{\delta}_3} = -\frac{\gamma}{N}\sum_{i=1}^{N}\ln\left(\frac{D - g_{3,i}}{D - T_3}\right) \tag{18}$$

These three estimators are numerically equal. Indeed, using $g_{1,i} = -\ln r_i$, $g_{2,i} = r_i^{-1/\alpha}$ and $g_{3,i} = K - r_i^{1/\gamma}$, one gets

$$\frac{1}{\hat{\delta}_1} = \frac{1}{N} \sum_i (-\ln r_i + \ln R) = -\frac{1}{N} \sum_i \ln \frac{r_i}{R} \tag{19}$$

$$\frac{1}{\hat{\delta}_2} = \frac{\alpha}{N} \sum_i \ln \left( \frac{r_i}{R} \right)^{-1/\alpha} = -\frac{1}{N} \sum_i \ln \frac{r_i}{R} \tag{20}$$

$$\frac{1}{\hat{\delta}_3} = -\frac{\gamma}{N} \sum_i \ln \left( \frac{r_i}{R} \right)^{1/\gamma} = -\frac{1}{N} \sum_i \ln \frac{r_i}{R} \tag{21}$$

The estimated inverse dimensions $1/\hat{\delta}_a$ are all equal to

$$\frac{1}{\hat{\delta}_a} = -\frac{1}{N} \sum_i^N \ln \frac{r_i}{R} = -\ln \mathrm{geom} \left( \frac{r_1}{R}, \dots, \frac{r_N}{R} \right) \tag{22}$$

which is minus the mean of $\frac{r_i}{R}$ in logarithmic scale, or minus the logarithm of the geometric mean of the ratios $\frac{r_i}{R}$. Since the $\hat{\delta}_a$'s are all the same, we use from now on the notation $\hat{\delta}$ instead. This is precisely a local version of the expression of the estimator of the local dimension as given by Takens (Takens, 1985).

This ML estimator is particularly interesting in this case since it is, even for finite $N$, unbiased and efficient as estimator (James, 2006).

These three different ML principles are really equivalent since the loglikelihood functions are all equal up to a term independent of $\delta$ (so that the derivatives of the loglikelihood functions are equal). Indeed, using the different but equal expressions of $\hat{\delta}$ in terms of the $g_{a,i}$'s, one has

$$\ln L_1(\delta) = N \ln \delta - \delta \sum_i (g_{1,i} - T) = N \ln \delta - N \delta \hat{\delta}^{-1} \tag{23}$$

$$\ln L_2(\delta) = N \ln(\delta\alpha) - (\delta\alpha + 1) \sum_i \ln \frac{g_{2,i}}{T_2} = N \ln(\delta\alpha) - (\delta\alpha + 1) \frac{N}{\hat{\delta}\alpha} = \ln L_1(\delta) + N \ln \alpha - \frac{N}{\hat{\delta}\alpha} \tag{24}$$

$$\ln L_3(\delta) = N \ln(\delta\gamma) + (\delta\gamma - 1) \sum_i \ln \left( \frac{K - g_{3,i}}{K - T_3} \right) = N \ln(\delta\gamma) - (\delta\gamma - 1) \frac{N}{\hat{\delta}\gamma} = \ln L_1(\delta) + N \ln \gamma + \frac{N}{\hat{\delta}\gamma}. \tag{25}$$

As functions of $\delta$, the three different loglikelihood functions are thus equal, up to an additive term independent of $\delta$. This is why the $\hat{\delta}_a$'s have all the same value and why the confidence bounds based on the likelihood function do not depend on the chosen function $g_a(r)$ (see section (2.1.3)). In the following, we use only the function $g_1(r)$ unless explicitly stated.

In Pons et al. (2023), the authors also estimate the shape parameter ($\xi$) of the observed extreme value law of the $g_{a,i}$'s. We do not do it since it is the choice of the function $g_a(r)$ which fixes the extreme value law to expect. If there is a deviation from this law, it is because the scaling $C(R) \sim R^\delta$ is not satisfied in the first place (see section 2.5 for possible reasons why $C(R) \sim R^\delta$ does not hold).

### 2.1.3 Confidence intervals

Confidence bounds on the estimations of the dimension were already studied to some extent in Theiler (1990), but through the statistical error on the estimation of $C(R)$. Since we do not rely on estimations of $C(R)$ and use only a ML principle for $\hat{\delta}$, we present here the computation of the bounds of the confidence intervals of $\delta$ using the likelihood function.

The loglikelihood function in the first case ($g_1(r) = -\ln r$) is

$$\ln L_1(\delta) = N \ln \delta - N \delta \hat{\delta}^{-1} \tag{26}$$

and $\ln L_2$ and $\ln L_3$ are the same functions of $\delta$, up to an irrelevant term, as shown in section 2.1.2. We have thus

$$\Delta \ln L_1(\delta) = \Delta \ln L_2(\delta) = \Delta \ln L_3(\delta) \tag{27}$$

where $\Delta \ln L_a(\delta) = \ln L_a(\delta) - \ln L_a(\hat{\delta})$. Since the computation of confidence bounds using the likelihood involves only $\Delta \ln L$, the confidence bounds will be the same for all three cases.

Given $N$, one can have a $n_\sigma$ confidence interval by finding the values of $\delta$ for which (see left panel of Fig. 2, James (2006))

$$\Delta \ln L(\delta) = N \left( \ln \frac{\delta}{\hat{\delta}} - \frac{\delta}{\hat{\delta}} + 1 \right) = -\frac{n_\sigma^2}{2} \tag{28}$$

Vice versa, if we fix some target confidence interval $[\delta_{min}, \delta_{max}]$ around $\hat{\delta}$, we can look for the minimum value of $N$ to use. The above equation turns into an inequality, of which there are two versions (for $\delta_{min}$ and $\delta_{max}$). Taking into account the most restrictive one, one has

$$N \geq \max \left[ N_{n_\sigma} \left( \frac{\delta_{min}}{\hat{\delta}} \right), N_{n_\sigma} \left( \frac{\delta_{max}}{\hat{\delta}} \right) \right], \quad N_{n_\sigma} \left( \frac{\delta}{\hat{\delta}} \right) = -\frac{n_\sigma^2}{2} \left( \ln \frac{\delta}{\hat{\delta}} - \frac{\delta}{\hat{\delta}} + 1 \right)^{-1} \tag{29}$$

For example, if we want a 10% interval (i.e. $\delta_{min} = 0.9\hat{\delta}$ and $\delta_{max} = 1.1\hat{\delta}$) with 95.5% confidence ($n_\sigma = 2$), the required $N$ is computed as

$$N \geq \max(N_2(0.9), N_2(1.1)) = \max(373.1, 426.5) = 426.5. \tag{30}$$

So $N$ has to be bigger than 427 to be sure at 95.5% that the true dimension is at most 10% below or 10% higher than $\hat{\delta}$.

The right panel of Fig. 2 shows a plot of the functions $N_2 \left( \frac{\delta_{min}}{\hat{\delta}} \right)$ and $N_2 \left( \frac{\delta_{max}}{\hat{\delta}} \right)$.

One can also invert the relationship to compute $\frac{\delta min}{\hat{\delta}}$ and $\frac{\delta max}{\hat{\delta}}$ in terms of $N$:

$$\frac{\delta_{min}}{\hat{\delta}} = -W_0(-e^{-a}), \quad \frac{\delta_{max}}{\hat{\delta}} = -W_{-1}(-e^{-a}) \tag{31}$$

where $W_0$ and $W_{-1}$ are the Lambert function or order $0$ and $-1$ respectively, and $a = \frac{n_\sigma^2}{2N} + 1$. Note that these relative bounds (i.e. the bounds for $\delta$ relatively to $\hat{\delta}$) depend only on the number of observations, i.e. on the number of analogues, and not on the dimension itself!

These bounds should be taken with caution. They are computed supposing that the underlying distribution of points is indeed exponential and that all the samples are IID, as was already pointed out in Theiler (1990). In the case of a dynamical system

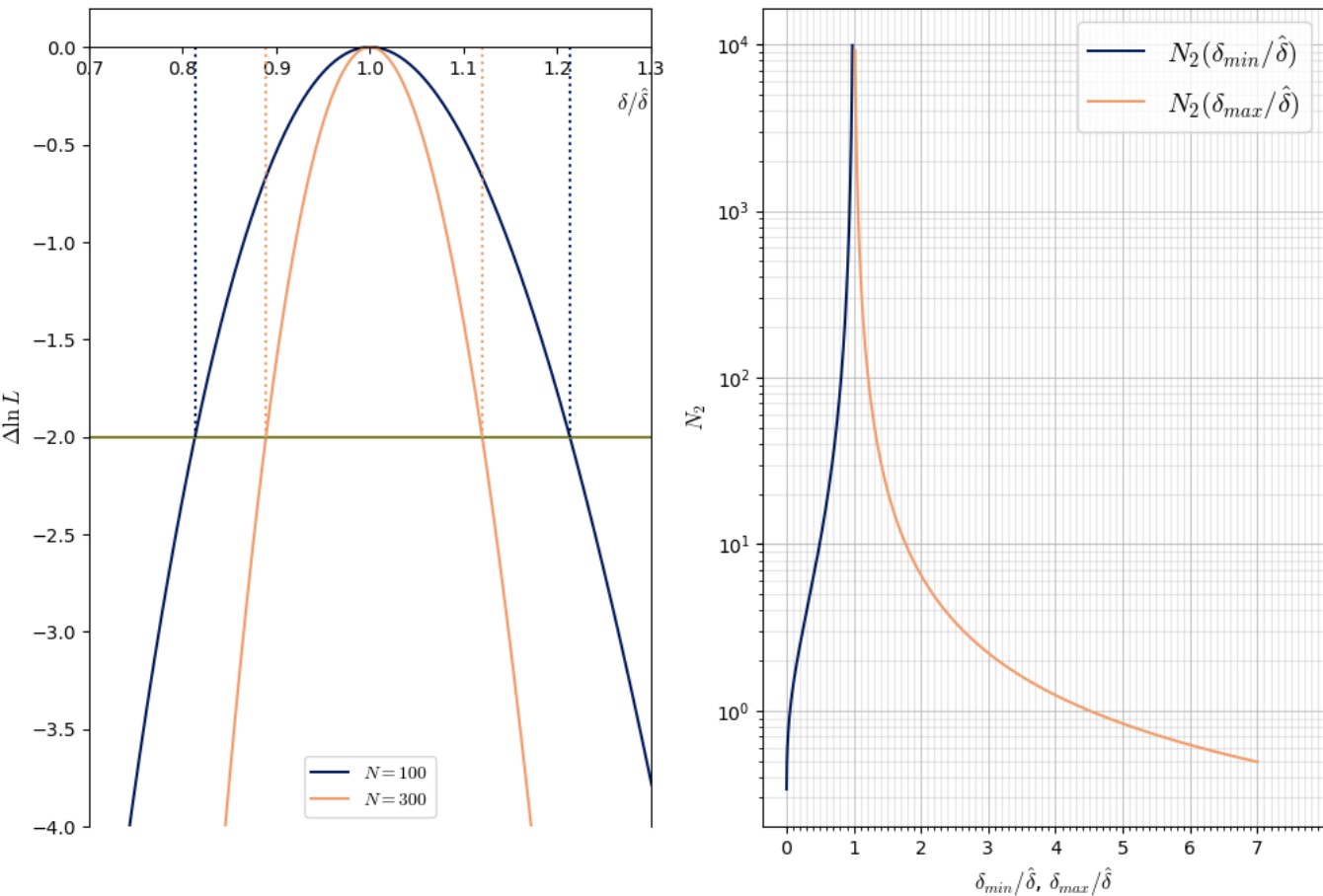

**Figure 2.** Left: Plot of $\Delta \ln L$ as a function of $\frac{\delta}{\hat{\delta}}$ for $N = 100$ and $N = 300$. The green horizontal line corresponds to $\Delta \ln L = -\frac{n_\sigma^2}{2}$ ($n_\sigma = 2$). The dotted lines show how to read $\frac{\delta_{min}}{\hat{\delta}}$ and $\frac{\delta_{max}}{\hat{\delta}}$ for the two different values of $N$. Right: Plot of $N_2$ versus of $\frac{\delta_{min}}{\hat{\delta}}$ and $\frac{\delta_{max}}{\hat{\delta}}$ ($n_\sigma = 2$, 95.5% confidence).

producing the dataset, this is only true in the asymptotic limit. Consider for example a system observed with very high time resolution but on a quite short time. In this case, the system could have visited only a few times the ball $B(R)$ (so that the system has not explored all directions around $\zeta$ yet) but one could still have $N \geq 427$. In such case, the distribution of values of $r$ around $\zeta$ has not converged yet and we cannot really trust the confidence interval $0.9\hat{\delta} \leq \delta \leq 1.1\hat{\delta}$. It is however unlikely that the observed distribution of values of $r$ looks like an exponential in that case. The NRMSE score introduced below helps to quantify how the experimental distribution is far from an exponential distribution.

## 2.2 NRMSE score

The ML method produces an estimate of the dimension even if the distribution of points is not at all an exponential, in which case the estimated dimension and its bounds are not correct. A systematic way to assess whether the fitted distribution is indeed exponential or not is therefore needed.

As in Datseris et al. (2023), we tried to use a Kolmogorov-Smirnov test but it did not prove to be very efficient to assess the quality of the exponential fit to the data. Following their suggestion, the normalized root mean squared error (NRMSE) is used instead between the fitted version of the exponential distribution $f_{1,T_1}$ and a uniform distribution. It is computed as follows:

1. the data are first binned in bins of equal size and the empirical probability $P_i$ associated with the bin $i$ is computed as the fraction of observations lying in this bin (the number of bins is taken as the minimum between the Sturge's rule and the Freedman-Diaconis rule, see Scott (2015) about those rules);

2. the $E_i$'s are defined as the fraction of events that would fall into each of the bins if the events followed an exponential law characterized by $\hat{\delta}$;

3. $U$ is defined as the probability of falling into each bin using a uniform law over the range of observations (it does not depend on the bin because they are taken to have all the same width);

4. the NRMSE score is then computed as

$$\text{NRMSE} = \sqrt{\frac{\sum_i (P_i - E_i)^2}{\sum_i (P_i - U)^2}} \tag{32}$$

The NRMSE score gives an indication of how much the data is better described by an exponential law than by a uniform law. For a good fit, we expect a small NRMSE score: in this case, the numerator (which is the error between the experimental distribution and the fitted exponential distribution) is much lower than the denominator (which is the error between the experimental distribution and the uniform distribution).

## 2.3 Interpretation of the estimated dimension

The interpretation of the dimension is more easily understandable in the third case with $\gamma = 1$ and $K = 0$: the observable is in this case $g_3(x) = -x$. Since the values of $\delta$ obtained in each case are equal, the interpretation holds irrespectively of the function $g_a$ used.

Figure 3 shows two possible histograms for two random variables following both the distribution $f_{3,T_3} = \left(\frac{g_3}{T_3}\right)^{\delta-1} = \left(\frac{r}{R}\right)^{\delta-1}$ (i.e. with $K = 0$ and $\gamma = 1$), but for two different values of $\delta$. If the empirical distribution of radii $r_i$ (normalized by $R$) looks like that of the orange histogram, the estimated value $\hat{\delta}$ will be lower than if the empirical distribution is closer to the dark blue histogram. Mathematically, this is because the higher the value of $\delta$, the steeper $r^{\delta-1}$ is. This means that, the higher the dimension, the more the analogues inside $B(R)$ will be close to the boundary of $B(R)$. In the following, the estimated value of the dimension is sometimes artificially high because of that.

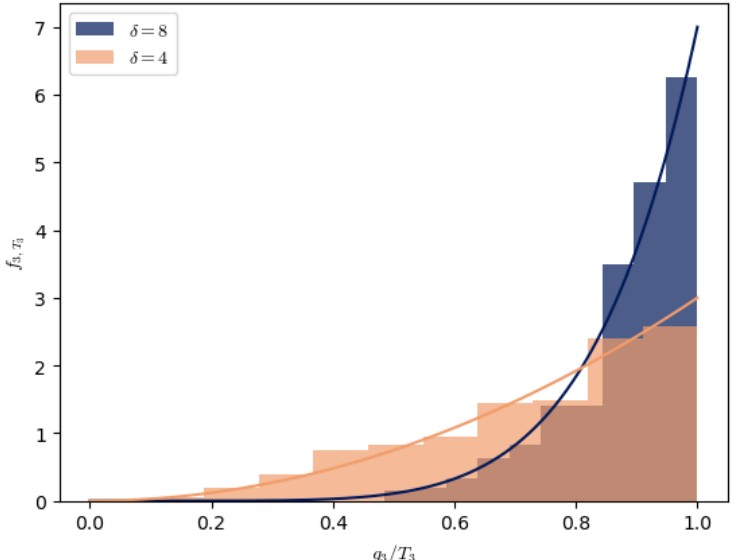

**Figure 3.** Example of histograms of two random variables $g_3(r) = -r$, whose PDFs for both are $f_{3,T_3} = \left(\frac{g_3}{T_3}\right)^{\delta-1} = \left(\frac{r}{R}\right)^{\delta-1}$ (i.e. with $D = 0$ and $\gamma = 1$), but for different values of $\delta$. The horizontal axis is $\frac{g_3}{T_3} = \frac{r}{R}$.

## 2.4 Measure a high dimension with a small $N$?

The computation of section 2.1.3 indicates that with $N \gtrsim 400$ (and if the NRMSE score is good), a $10\%$ accuracy is reached for the estimation of $\delta$, for all values of the dimension. This seems in contradiction with the argument of Eckmann and Ruelle (1992) (i.e. the number of points in $B(R)$ should be an exponential of $\delta$). In this section, we examine in detail how the scaling $C(R) \sim R^\delta$ is consistent with the computation in 2.1.3.

We set ourselves again in the first case, using $g_1(r) = -\ln r$ as observable. If we denote by $x_i$ the combination $\ln \frac{R}{r_i}$ (also equal to $g_{1,i} - T_1$), the computation of section 2.1.1 implies that the $x_i$'s follow an exponential law $f_1(x) = \delta e^{-\delta x}$. In terms of this exponential distribution, $\delta^{-1}$ is interpreted as the scale (see Fig. 4). The ML estimator of the scale of an exponential is the mean of the $x_i$'s, so that we recover the expression $\hat{\delta}^{-1} = \frac{1}{N}\sum_i x_i$ from section 2.1.2. From this point of view, a large dimension just means a smaller scale for the exponential, which is why it is not much more difficult to measure.

Let us write $r_0$ for the smallest of the $r_i$'s and $x_0 = \ln \frac{R}{r_0}$ for the biggest of the $x_i$'s. One has obviously $\delta^{-1} < x_0$. If there are enough points to estimate $\delta^{-1}$, $x_0$ must be quite far to the right of the plot in Fig. 4, so that a necessary condition is $\delta^{-1} \ll x_0$, or

$$s \gg 1, \quad s \equiv x_0\,\delta \tag{33}$$

This expresses that $x_0$ has to be large with respect to the scale $\delta^{-1}$ of the exponential. Conversely, if $s \gg 1$, this means that the exponential has been sampled enough to get a high value for $x$ (namely $x_0$).

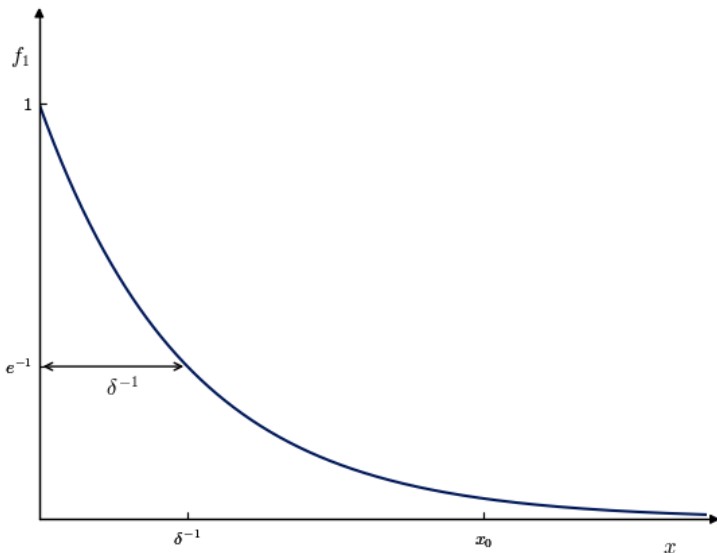

**Figure 4.** An exponential distribution and its scale parametrized by $\delta^{-1}$.

One could think that the condition (33) is actually easier to fulfill if the scale $\delta^{-1}$ is smaller (and the dimension bigger), because this leaves "more room" for $x_0$. But this is not the case, since a smaller scale means that the exponential will be more peaked near $x = 0$, so it will be more difficult to have a sample $x$ with a high value. This is consistent with the fact that the number $N = C(R)$ of points inside $B(R)$ satisfies

$$\frac{R}{r_0} = N^{1/\delta}, \tag{34}$$

(using the scaling $C(R) \sim R^\delta$ and the fact that $C(r_0) = 1$ since $r_0$ is the smallest of the $r_i$'s). This equality shows that, for a fixed $N$, the ratio over which the scaling (3) holds quickly approaches 1 when $\delta$ increases. This is the price to pay to measure high dimensions with a reduced number of points, and this poses some difficulties when dealing with large systems (see section 4).

Some remarks:

– Using (34), one can see that $s = x_0 \delta$ should be just $\ln N$ and the condition (33) is simply

$$\ln N \gg 1. \tag{35}$$

In practice, the distributions are exactly exponential only in the limit $R \to 0$, so that $s$ and $\ln N$ are not exactly equal.

– This equivalence between the estimation of different dimensions is because of the scaling transformation of the exponential distribution $f_1$:

$$\begin{cases} x & \to & \lambda x \\ \delta & \to & \frac{\delta}{\lambda} \end{cases} , \quad f_1(x) = \delta e^{-\delta x} \to \frac{1}{\lambda} f_1(x) \tag{36}$$

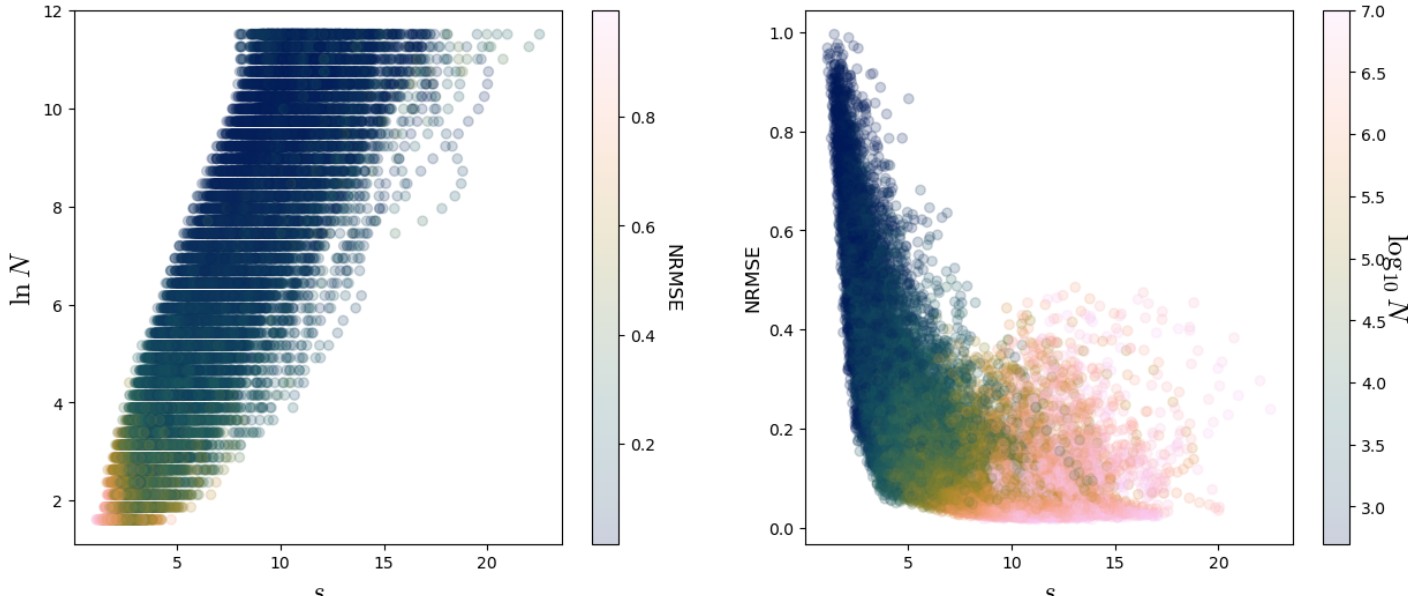

**Figure 5.** For the Lorenz 63 system (see Appendix A) with $\rho = 28$, $10^6$ points. Left: scatter plot of $\ln N$ against $s$. The color is the NRMSE of the fit. Right : scatter plot of the NRMSE against $s$. The color is the $10-$logarithm of the number of points used in the exponential fit. There are 1000 points and 20 fits for each.

which makes the difficulty of measuring a high value of the dimension similar to the difficulty of measuring a small value. Again, in terms of an exponential distribution, the dimension is just the inverse of the scale of the exponential. The probability density functions $f_{2,T_2}$ and $f_{3,T_3}$, and adapted definitions of $x$ in each case, have similar scaling properties. These make possible to absorb a redefinition of $\delta$ in a redefinition of $x$ if one uses $f_{2,T_2}$ or $f_{3,T_3}$ instead of $f_{1,T_1}$.

– If the EVT estimator for $\delta$ is able to produce a high value over some range, the estimation of the dimension through the correlation dimension can also do it. Indeed, in both cases, $\hat{\delta}$ is obtained by somehow fitting the scaling $C(R) \sim R^\delta$. The limitation for the correlation dimension is the same as for the EVT dimension: when $\delta$ increases, the range over which this scaling can hold decreases.

     The left panel of Fig. 5 shows a scatter plot of $\ln N$ against $s$, with the color being the NRMSE score for dimensions in
the Lorenz 63 system (see Appendix A about the Lorenz 63 system). Each point in both panels of Fig. 5 represents a fit. We computed the dimension for 1000 points, for 20 different values of $N$, so there are 20 000 points in each scatter plot of this Figure. A given computation point $\zeta$ in the phase space is therefore represented several times. One can see a rough agreement between $\ln N$ and $s$. Here and in the following, we computed $s$ as $s = x_0 \hat{\delta}$.

     The right panel of Fig. 5 shows a scatter plot between the NRMSE score and $s$ for the Lorenz 63 system. The form of this
plot is quite characteristic and will be encountered several times in the rest of the paper. The NRMSE score clearly decreases as $s$ increases: this part corresponds to fits increasingly better. When $s > 10$, the NRMSE reaches a plateau. The NRMSE for

some high values of $s$ and $N$ is not so good, this is when the $B(R)$ ball is too big and the distribution in it cannot be a power law (see section 3.2). In practice, one finds that $s$ bigger than $4-5$ seems to give NRMSE scores below $0.4$. Reliable fits can be selected by keeping for example only those for which $s > 5$ and NRMSE$< 0.4$.

In summary, the section 2.1.3 has shown that, for a given accuracy on $\delta$ relatively to $\hat{\delta}$, all dimensions require the same number of points inside $B(R)$. This might seem surprising because of the scaling $C(R) \sim R^\delta$, but can actually be understood if we see $\delta^{-1}$ as the scale of the exponential distribution of the $g_1$'s. However, the range of values of $R$ over which the dimension can be measured decreases with the dimension. This is why there is no contradiction with the original argument of Eckmann and Ruelle (1992), which supposed that $R$ is fixed.

In the following analyses, we use scatter plots in the form of that of the right panel of Fig. 5. The rough agreement between $s$ and $\ln N$ can be seen as a consistency check that the distributions of values of $g_{1,i}$ are indeed close to exponentials (and equivalently, that the distributions of $r_i$ are close to power laws).

## 2.5    Possible phenomena affecting the value of the dimension

The last sections developed tools to quantify how good is a fit to the exponential law. However, even if the fit is good, different
phenomena can affect $\hat{\delta}$ for a given $R$.

If the NRMSE or $s$ do not have good scores, the points in $B(R)$ do not follow a $R^\delta$ law. The cause of this could be one of the following:

1. There are not enough points inside $B(R)$ to properly recognize an exponential (and make the difference with a uniform law for example).

2. The points are not statistically independent, as in the example mentioned at the end of section 2.1.3. This was already noted in Theiler (1990).

3. $R$ is too big. To observe the $C(R) \sim R^\delta$ law, one indeed needs to consider a flat neighborhood around $\zeta$ and, if $R$ is too large, this law could be affected by the curvature (Perinelli et al., 2023), or by another geometric feature entering $B(R)$. See section 3.2 and Fig. 8 for an example of that.

On the other hand, if the NRMSE and $s$ give good scores, this means that the number of points inside $B(R)$ indeed follows a power law $R^{\hat{\delta}}$. This could have different causes and a non-exhaustive list is:

1. The estimated value of the dimension can indeed reflect the dimension of the surface supporting the neighboring points. This happens if the density of points is constant and the "surface" of the attractor is approximately flat.

2. If there is a clear structure of points in the phase space, with the density of points of this structure being higher enough
than its surroundings, the estimated dimension $\hat{\delta}$ will be that of this structure. This is because the intersection of such a structure with the ball $B(R)$ will make the number of points scale as $R^\delta$, where $\delta$ is the dimension of the structure. Such a situation is shown schematically in Fig. 6: in a plane, there is a straight line crossing $B(R)$ and the density $\lambda$ of points on

that line is much higher than anywhere else in $B(R)$. The number of points in $B(R)$ will then be essentially the number of points on that line. A simple computation taking only into account the points of the line gives $C(R) = 2\lambda\sqrt{R^2 - a^2}$, and if $a \ll R$ (i.e. if $\zeta$ is close enough to the line), $C(R) \approx 2\lambda R$. The scaling of $C(R)$ around $\zeta$ becomes that of the line, even though $\zeta$ is not strictly on it.

The previous case and this one are cases where the estimated dimension is the dimension of some geometric structure in the phase space. This geometric dimension is only well-defined for ranges of $R$ where the geometry is homogeneous in this range: if the curvature changes or a different geometric feature enters $B(R)$ for some value of $R$, one can not give a geometric meaning to the dimension. It is only for the values of $R$ for which there are clear objects that the dimension can be interpreted geometrically. In particular, one can approach the pointwise dimension in the limit $R \to 0$ only when $R$ is below any other geometric scale. This will be illustrated in the next sections.

Note that this kind of geometric dimension is conceptually equivalent to the one measured by local PCA techniques (such as in Little et al. (2017)).

3. The fact that $C(R) \sim R^\delta$ supposes that the density of points is uniform in the range over which the estimation is done. If the density is increasing when getting away from $\zeta$ (i.e. for increasing $r$, $\Sigma \sim r^a$ for $a > 0$), the empirical histogram of the values of $r$ will be inflated for values of $r$ close to $R$. In that case, the estimated distribution will be closer to the dark blue curve than to the orange curve in Fig. 3, so that the estimated dimension will be higher than the dimension of the surface supporting the points (the "geometric dimension"). In the same way, if the density of points decreases, the estimated dimension will be lower (see remarks in section 2).

In practice, it is not likely that the density $\Sigma$ will follow the same behavior $\Sigma \sim r^a$ on a large range of values of $r$. The fit in this case is usually not so good, so the selection using the NRMSE and $s$ (see section 2.4) should discard some of them. One can also try to visually identify when this happens because the dimension is fluctuating a lot.

Unfortunately, the tools we developed (the NRMSE and the quantity $s$) can produce good scores in all three above cases and do not allow to make the difference between each of these cases. One has therefore to keep in mind all possible phenomena affecting the estimation of the dimension, and inspect the values of the dimension over some range of values of $R$, in order to interpret properly the geometry of the system in this range.

In the following, as is usual when computing dimensions from time series, a Theiler window around each computation point $\zeta$ was applied (Theiler, 1986), in order to avoid having points too close in time inside the ball $B(R)$.

## 3  Application to small systems

To illustrate some aspects described in the previous sections, we compute the local dimension for small systems, allowing to get some insight of what is captured by the estimated dimension.

The classical Lorenz 63 system for $\rho = 28$ (Lorenz, 1963) is first considered. The relatively simple geometry allows to illustrate some phenomena described in section 2.5. We then present the results of the computation of the local dimension for

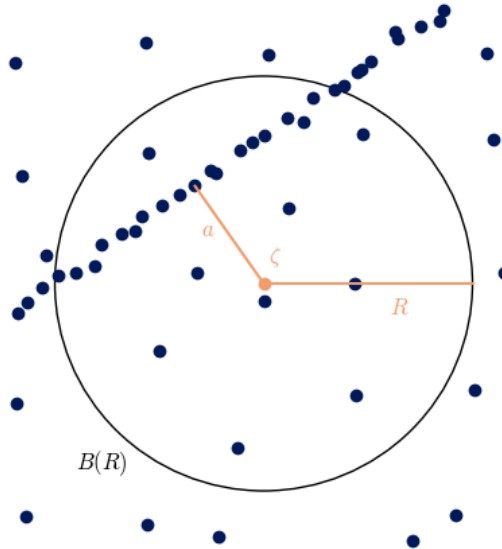

**Figure 6.** Picture of a situation where almost all points on the attractor (in dark blue) lie on a line. If the density of points is constant and denoted $\lambda$, the number of points inside $B(R)$ is essentially $2\lambda\sqrt{R^2 - a^2}$. For $a \ll R$, the estimated dimension will be 1.

the Lorenz 63 system with $\rho = 166.5$, which is intermittent (Pomeau and Manneville, 1980; Sparrow, 1982). In the case of intermittent systems, a situation like the one displayed in Fig. 6 is often encountered, and the tools described in section 2 can be used to detect structures in phase space.

### 3.1 How we choose the values of $R$

If we want to compute the dimension for different computation points $\zeta$, it is difficult to chose relevant values of $R$ for each of them if we do not know very well the system. Indeed, depending on the dimension and on the density of points, the values of $R$ needed for $B(R)$ to contain a sufficient number of points to estimate $\delta$, might be different for each computation point $\zeta$.

An alternative approach is to compute the distance between the computation points and all other points of the trajectory and keep a given percentage $q$ of points (i.e. consider as extremes), common to all computation points. For each computation point $\zeta$, $R$ is then defined such that $B(R)$ contains $q\%$ of the total number of points in the dataset (i.e. $R$ is the $q^{th}$ percentile). For each computation point and for each percentage $q$ for which we compute the dimension, there is thus one value of $R$. This has the advantage to adapt the range over which the dimension is computed accordingly to the computation point.

Of course, the percentile will not be exactly the same if one increases the length of the series. However, if the attractor has been reasonably sampled by $N_{tot}$ points, additional points should spread in the phase space in the same proportions, so that the percentiles computed with $N_{tot}$ or $N'_{tot} > N_{tot}$ should correspond. In other words, if the density of points is already close to

405 the invariant measure of the attractor $\mu$, the computed percentiles will not significantly change when the length of the trajectory is increased.

In the following sections, we will use this method with percentages chosen evenly spaced on a logarithmic scale, typically with the maximum percentage being $10\%$ and the smallest percentage corresponding to 5 analogues. The dimension computed with 5 points will not give a precise value (see section 2.1.3) but this allows to ensure to have a dimension computed over a

410 sufficiently large range of values of $R$. The values of the dimension with not enough points will be filtered out through the computation of $s$ and of the NRMSE score.

In section 3.3, we will also choose the values of the radius $R$ rather than computing it as percentiles, because it will be easier to illustrate the interpretation of the dimension in this case.

## 3.2 Lorenz 63 with $\rho = 28$

We start with the usual Lorenz 63 system with $\rho = 28$ (see Appendix A) and we consider two different trajectories: one with $N_{tot} = 10^4$ points, and the other one with $N_{tot} = 10^7$ points. Figure 7 shows the repartition of the dimension on the attractor. The dimension is computed with $q = 10\%$ (top, A and B) and with $q = 1\%$ (bottom, C and D), for these two trajectories. For $q = 10\%$, the two plots look the same and the dimensions agree between the two datasets. This is because $10\%$ of $10^4$ is already enough points to estimate the dimension, and adding more points to the dataset will not change the percentiles, nor change

the distribution of points on the attractor. Note that the points "on the border" of the attractor have higher dimensions since, from their point of view, the density of points is increasing. The estimated dimension for these points is actually an effective dimension reflecting this feature (see section 2.5).

On the other hand, for $q = 1\%$, there are not enough points in the $N_{tot} = 10^4$ dataset (Fig. 7 C) to have a proper estimation of the dimension, while the $N_{tot} = 10^7$ dataset (Fig. 7 D) still allows a proper estimation of the dimension. The latter gives

more homogeneous values of the dimension than in the corresponding plot (7 B) with $10\%$ of the points. This is because the values of the percentiles are smaller, so that the estimation is more local and the density less varying in the balls $B(R)$.

It is also interesting to analyze the histograms of the dimension for the $10^7$ points dataset, for $q = 1\%$ and $q = 10\%$ (Fig. 8, top): a lot of points have a dimension close to 1.5 for $q = 10\%$, but most points have a dimension close to 2 for $q = 1\%$. This is because, in the $q = 10\%$ case, $R$ is too large for some points: $2R$ is larger than the width of the wings of the attractor, so that

the $C(R) \sim R^2$ cannot hold. The two bottom scatter plots of Fig. 8 illustrate this. In the $q = 1\%$ case, $R$ is smaller and this never happens.

It is likely that, when decreasing $q$ and increasing $N_{tot}$ again in order to keep $N = qN_{tot}$ big enough, the dimension for all points will tend to a common value. This seems to be in agreement with the mentioned asymptotic results that the dimension of almost all points converge towards a unique value.

Indeed, as observed on Fig. 7, the dimension of the points on the wings seem to converge to some value close to 2, as expected for a surface. As $q$ decreases and $N_{tot}$ increases, the dimension of the points on the "border" of the attractor will behave as others points on the wings and their dimension would converge to the same value close to 2.

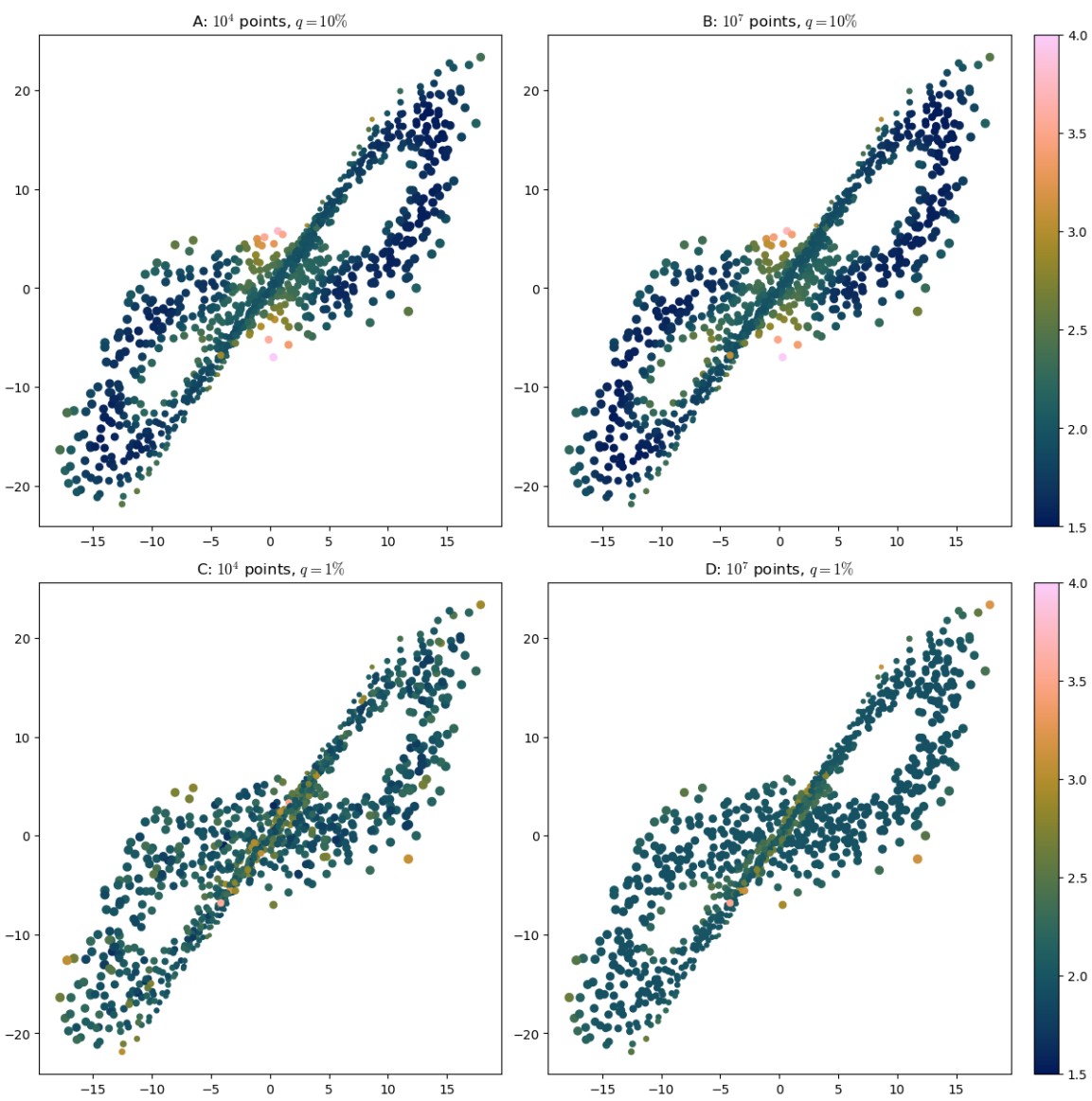

**Figure 7.** The Lorenz 63 attractor, with the dimension represented in color, computed with $10\%$ (top) and $1\%$ (bottom) of the points in each case (only the 1000 points for which the dimension is computed are represented).

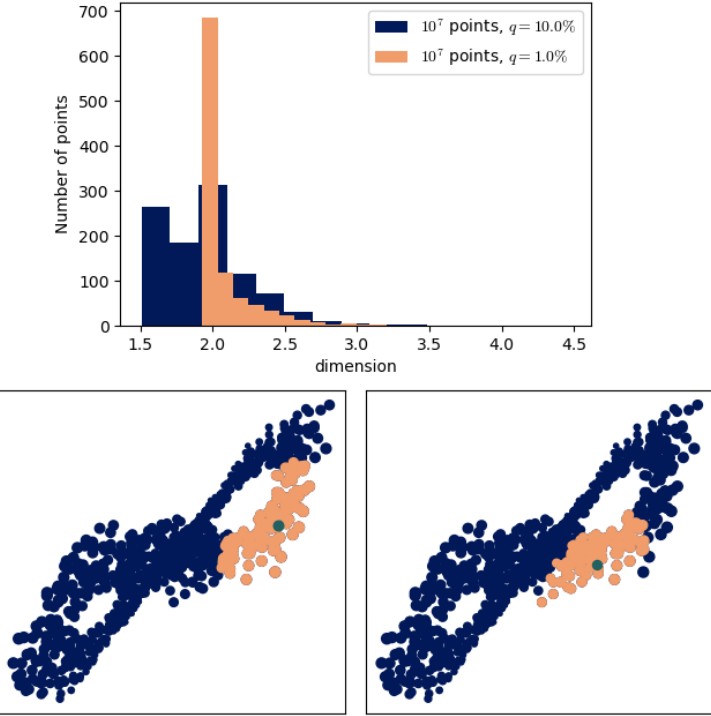

**Figure 8.** Top: Histograms of the dimension for the $10^7$ points dataset, for $q = 1\%$ and $q = 10\%$ (1000 values in each case). Bottom: Two illustrations that $R$ can be too large for $q = 10\%$. The computation point $\zeta$ is in green and some of the points used to compute the dimension are in orange. The ball $B(R)$ is bigger than the width of the wings, so that the $C(R) \sim R^2$ characteristic of a surface cannot hold.

The points close to the intersection of the wings have a dimension bigger than 2 as long as their balls $B(R)$ include this intersection. However, when $q$ is decreased, the balls $B(R)$ might not anymore enclose the intersection. The neighborhoods of those points then look as that of any other points on the wings, so that their dimensions will be close to 2. Only the points exactly at the instersection will always have a $B(R)$ including this intersection, so that their dimension will never approach 2.

The left plot of Fig. 9 displays the dimension estimate $\hat{\delta}$ as a function of $R$. There is a range of values for which the $\hat{\delta}$ is close to 2 for all points. The curves having a bump in the middle of the range of values of $R$ correspond to points near the intersection of the wings.

As one can see in the right panel of Fig. 9, some points with a high value of $s$ and a high number of points used in the estimation of $\delta$, have a poor NRMSE score (i.e. up to $0.4$, while most of the fits with $s > 10$ have their NRMSE score below $0.2$). Those correspond to situations shown in the bottom plots of Fig. 8, where the fit is not so good anymore.

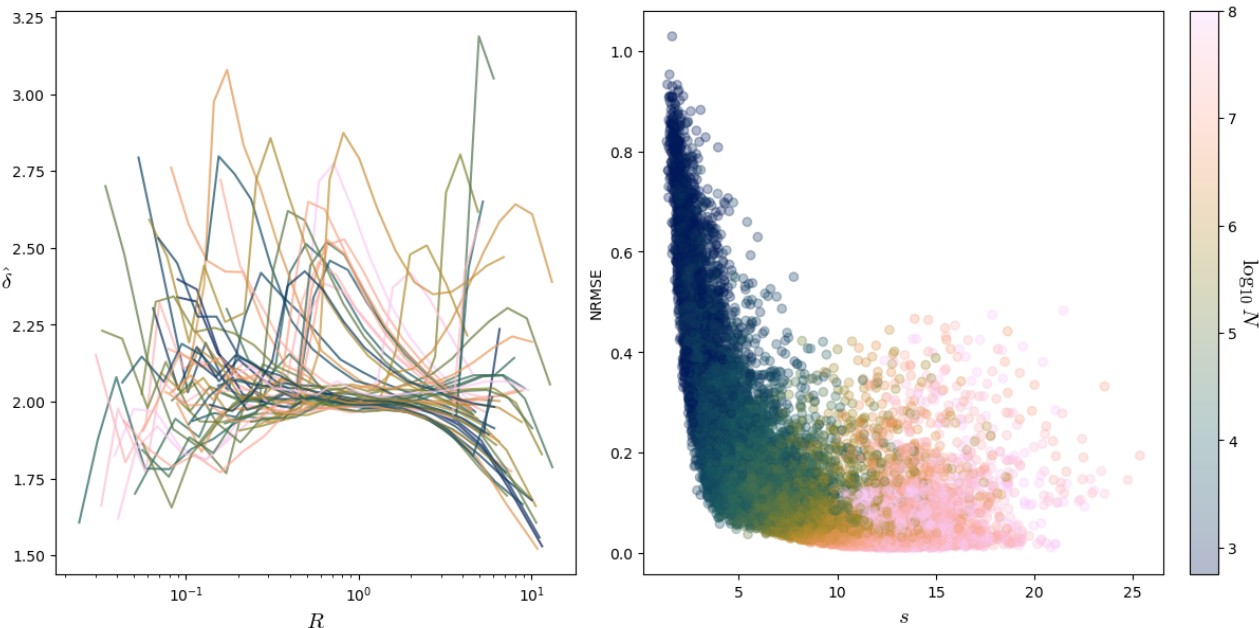

**Figure 9.** Left: Plot of $\hat{\delta}$ versus $R$ for 50 points of the Lorenz 63 system ($\rho = 28$) with a trajectory of $10^7$ points. Only fits with NRMSE$< 0.4$, $s > 5$ are plotted. Right: scatter plot of NRMSE vs $s$ for the same dataset, but for all 1000 computation points. The color represents the logarithm of the number of points used in the dimension estimation.

### 3.3 Lorenz 63 with $\rho = 166.5$

The results of the local dimension applied to the intermittent Lorenz 63 system with $\rho = 166.5$ are now analyzed (see Appendix A for a short presentation of this system). The case of intermittent systems is interesting since those systems have a strongly inhomogeneous phase space, allowing to illustrate how the dimension can have different values at different scales $R$. Intermittent systems are also of special interest, since the primary goal is to use the local dimension on the rain data, which is known to be intermittent.

This system was integrated for $\rho = 166.5$ to obtain $10^6$ points on the attractor and the dimension was computed for 1000 points. The plot of the dimension against $R$ (computed as percentiles) is in the left panel of Fig. 10. One can see that, for the smallest values of $R$, the dimension is around 2, while it is closer to 1 for the highest values of $R$.

Actually, the remaining of the attractor before the bifurcation (see Fig. A1) defines a $1-$dimensional structure in phase space. This structure is a closed loop and is made of the points in the laminar regime of this intermittent system. We expect that in the balls $B(R)$ intersecting enough this closed loop, the scaling will be strongly influenced by this closed loop (as described in section 2.5). A better insight of the behavior of the dimension with $R$ than that given by the left panel of Fig. 10, can be gained by characterizing each point by its position with respect to this closed loop.

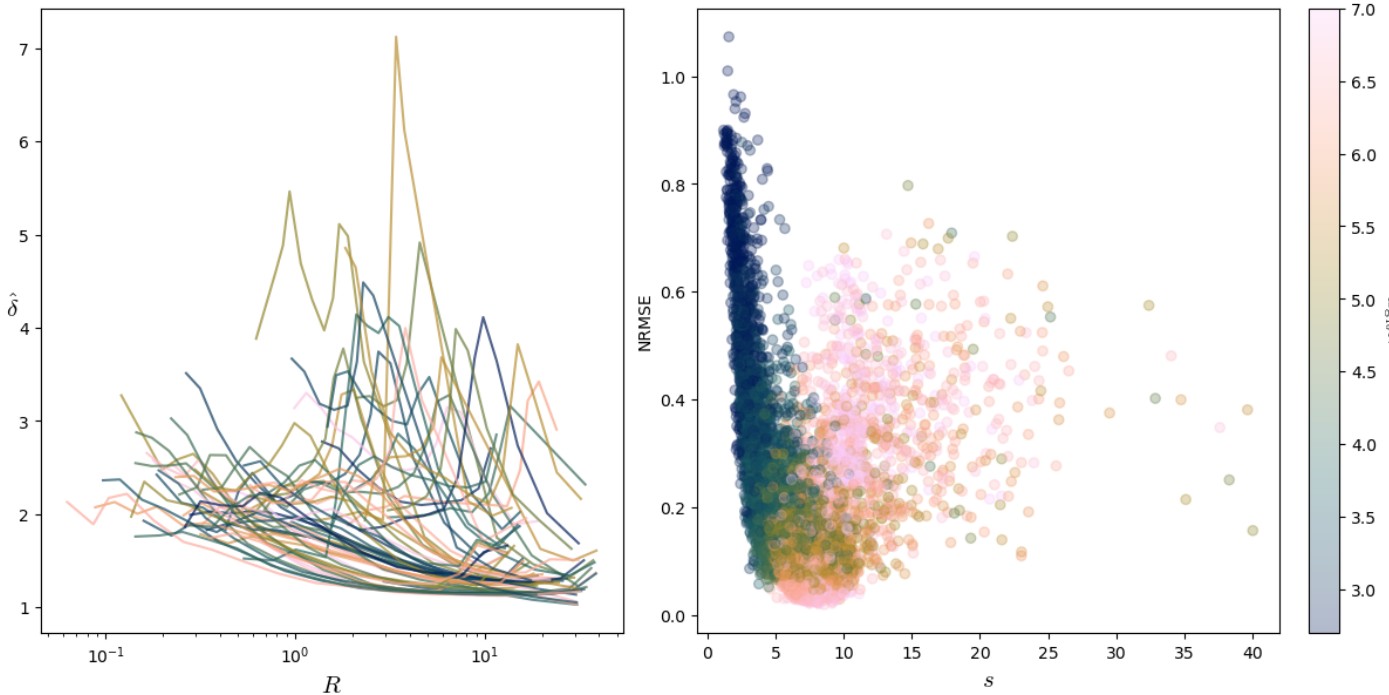

**Figure 10.** Left: estimated dimension $\hat{\delta}$ against $R$ for 100 points of the intermittent Lorenz 63 system ($\rho = 166.5$, trajectory of $10^6$ points). Only fits with NRMSE$< 0.4$ and $s > 5$ are shown. Right: scatter plot of NRMSE vs $s$ (all fits).

To do so, we select a part of the trajectory where it seems regular and almost periodic, and integrate this part of the trajectory with a very small time step. We then take this as a representation of the laminar regime. Of course, this part of the trajectory is not strictly periodic, so choosing different parts of the trajectory will lead to slightly different representations. This representa-
tion allows to define the "laminar distance" for any point $\zeta$ in the trajectory, as the smallest of the distances between $\zeta$ and all points in the representation of the laminar regime. From a dynamical point of view, points with small laminar distances will be considered as points in the laminar regime, while points with a significantly non-zero laminar distance can be considered as chaotic points.

Figure 11 shows the dimension as a function of the laminar distance for that point. More specifically, 6 values of the radius
$R$ ($R = 2, 5, 10, 15, 20, 30$) were chosen and the dimension for 1000 points is computed. Each panel corresponds to a specific radius $R$. In addition, the red vertical line marks where the laminar distance is equal to the radius $R$ used in this plot.

One can see from Fig. 11 that the dimension peaks for points whose laminar distance equals $R$. In other words, from the point of view of a given point:

    – when $R$ is smaller than the laminar distance, the dimension is around 2,

– when $R$ equals the laminar distance, the dimension has a peak,

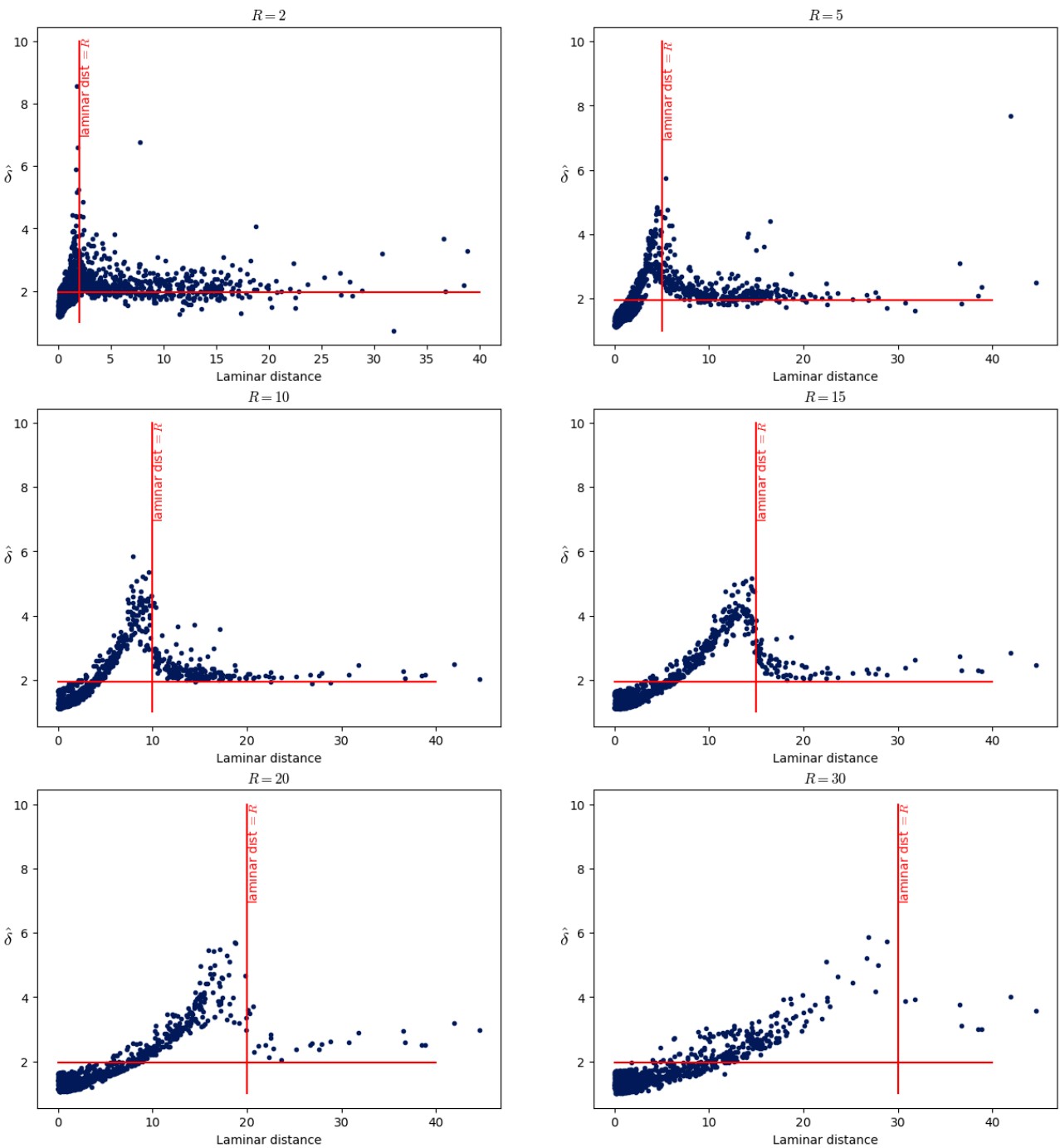

**Figure 11.** Scatter plots of the local dimension in terms of the laminar distance. In a plot, all the dimensions have been computed with the same radius $R$ (the value is above the plot). The red horizontal line marks the dimension value equal to 2, while the vertical red lines mark the value of the laminar distance which is equal to the radius $R$.

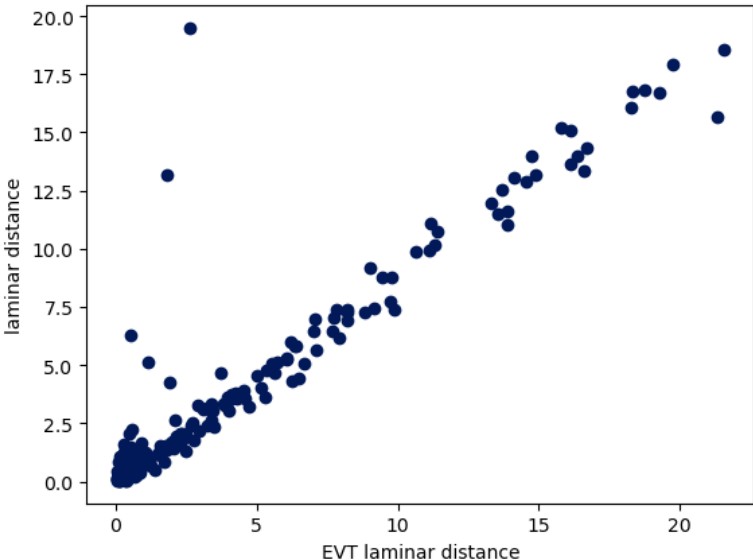

**Figure 12.** Comparison of the EVT laminar distance with the laminar distance for the intermittent Lorenz 63 ($\rho = 166.5$). The EVT laminar distance is computed as the distance for which the dimension is maximum (see text).

– when $R$ is bigger than the laminar distance, the dimension decreases between 1 and 2.

The fact that there is a peak in the dimension when $R$ equals the laminar distance can be understood by noting that it corresponds to the entrance of the laminar structure in the $B(R)$ ball around the computation point $\zeta$. There are suddenly a lot of points in the ball, near the boundary of the ball, so that the distribution of points in $B(R)$ resembles more the dark blue curve than to the orange curve in Fig. 3.

When $R$ is bigger than the laminar distance, the situation becomes similar to the one in Fig. 6 and the dimension becomes closer to 1. The dimension is 1 only if $R$ is sufficiently bigger than the laminar distance (i.e. $R \gg a$ in Fig. 6), which can happen only for points sufficiently close to the laminar regime. For other points, other parts of the loop forming the attractor in Fig. A1 would enter the $B(R)$ ball and modify the scaling, or the curvature effects become too important. In those cases (when $R$ is bigger than the laminar distance, but not much bigger), the distribution of points in $B(R)$ is not close to a power law, so that the exponential fit is not appropriate.

When computing the dimension in the asymptotic limit $R \to 0$, the radius $R$ should be below any other geometric scale around the computation point $\zeta$. For the chaotic points, this geometric scale is in this case their laminar distance, while for laminar points, the geometric scale is the "width" of the laminar structure in phase space.

As a consistency check for that interpretation, we tried to estimate the laminar distance for each of the 1000 computation points as the value of the radius $R$ for which the dimension is maximum. To achieve this, we used the values of the radius $R$ computed as percentiles and the corresponding dimensions. In order to use only meaningful values of the dimension, for each computation point, we restricted the dimension to the range of values of $R$ where NRMSE$< 0.5$ and $s > 4$. We rejected all

estimations where the maximum was found on one of the end of this range because this points to the fact that the true peak of the dimension is maybe outside of the range of values of $R$ we have for these points. Because of that, the laminar distance could be estimated for only one fourth of the 1000 computation points. The scatter plot in Fig. 12 shows the comparison of the laminar distance with this estimation of the laminar distance using EVT (which we call "EVT laminar distance").

Note that the EVT laminar distance tends to be bigger than the laminar distance itself. This is because, if $R$ is precisely equal to the laminar distance, there are not enough points in $B(R)$ for $\hat{\delta}$ to be really influenced. $R$ has to be a little bigger than the laminar distance for $\hat{\delta}$ to really increase, and the peak of $\hat{\delta}$ is for values of $R$ a little bigger than the laminar distance.

We carried out the same analysis on two other intermittent dynamical systems: the Lorenz 96 system with $n = 4$ variables for $F = 11.87$, and the Lorenz 96 system for $n = 12$ variables for $F = 4.4$. Figures similar to 11 for those two systems are displayed in Appendix B.

## 4 Application to large systems

In this section, we present the results of the computation of the dimension for two large systems: the Lorenz 96 system with $n = 50$ dimensions and the RADCLIM dataset (radar images of the precipitation field).

### 4.1 Lorenz 96, $n = 50$

The Lorenz 96 system (see Appendix B for a brief description) with $n = 50$ dimensions was integrated for $10^6$ time units, with a time step of $dt = 0.1$, for two values of the parameter: for $F = 4.9$ and for $F = 6$. Each of the two trajectories has $10^7$ points. The radii $R$ were computed as percentiles and the corresponding estimates of the dimension were computed in each case.

The results for $F = 6$ are presented in Fig. B3. Our method suggests that there are no salient geometric structure in phase space for that parameter value. We focus now on the $F = 4.9$ case.

The right plot of Fig. 13 shows as expected that the NRMSE score decreases when $s$ is increased. The left panel of Fig. 13 shows the estimated local dimension against the radius $R$ of the ball for 100 points in this system. We selected the points for which the NRMSE is smaller than $0.4$ and $s > 6$.

One can see at the bottom of the plot a set of curves having all a small maximum for radii between 8 and 12, and maximum dimension smaller than $\sim 12$ (dotted curves). These points seem to detect a structure of points at a distance $10 - 12$ from them. If there was such a structure at that distance, we would see other curves with maxima for small values of $R$ for the points which are part of this structure, as for the intermittent Lorenz 63 in section 3.3. Since there are no such curve, and because the dotted curves look quite regular, and also because they correspond to a great proportion of all the curves, we can think that the points corresponding to these curves are part of the detected structure. In other words, the dotted curves correspond to points which are part of the structure that these curves detect. Geometrically, the points of these curves have a similar role than the laminar points of the section 3.3.

To understand how a structure can be detected by the curves of the points which are part of that structure, consider points uniformly distributed on a circle of radius $\tilde{R}$. From the point of view of a point on the circle, the number of points at a distance

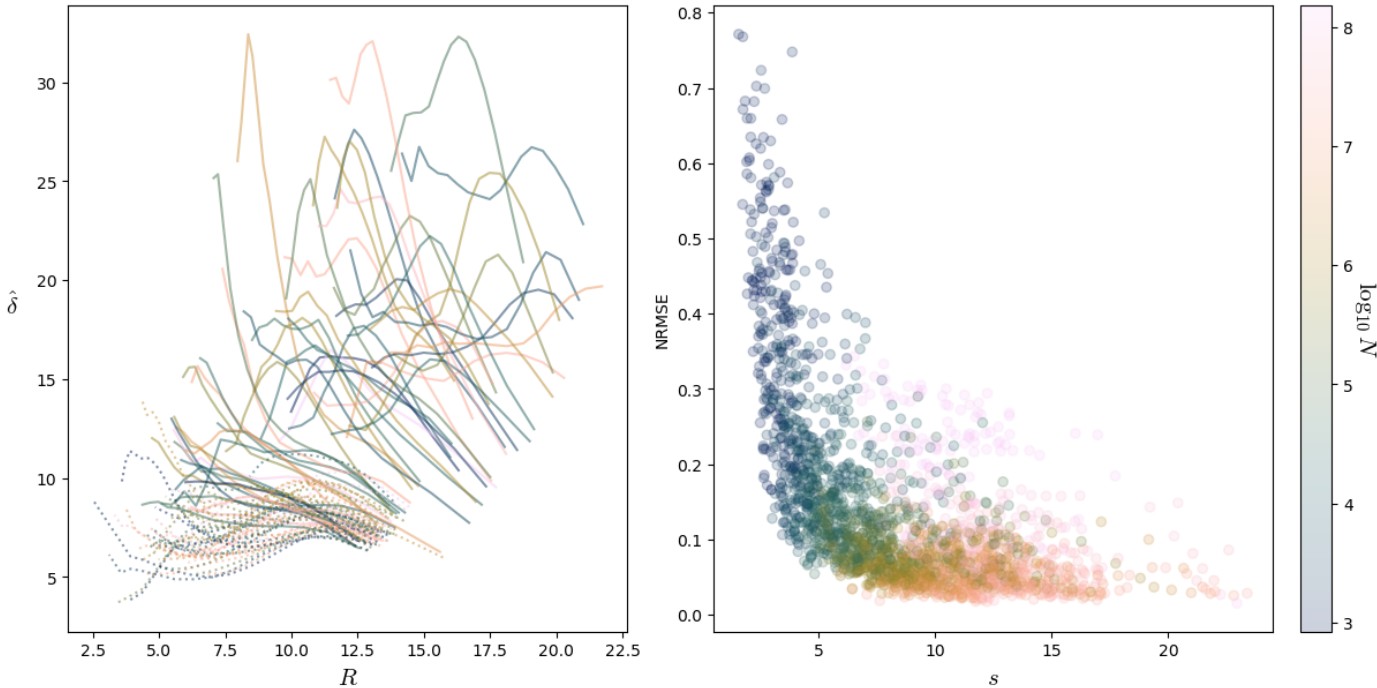

**Figure 13.** Left: Plot of the dimension against the radius $R$ for 100 computation points for Lorenz 96 with $n = 50$ and $F = 4.9$. Only fits for which NRMSE<0.4 and $s > 6$ are shown. The dotted lines detect a structure and the corresponding points are part of that structure. Right: Scatter plot of the NRMSE vs $s$ for the Lorenz 96 system with $n = 50$ and $F = 4.9$, with $\log_{10} N$ in color.

$< R$ grows as $\tilde{R} \arccos\left(1 - \frac{R^2}{2\tilde{R}^2}\right)$. This curve is the steepest when $R$ approaches $2\tilde{R}$, so that the fitted $\delta$ would have a peak for $R \sim 2\tilde{R}$. This peak is analogous to the peak of the dotted curves in the left plot of Fig. 13.

The curves which are not in this set are much more diverse. They all start at a higher value of the radius, which is because the corresponding points are in less dense parts of the phase space. Typically, their starting value of $\hat{\delta}$ is also higher, and this is because the density of points around those points typically increases. This leads to artificially high values of $\delta$ (see $\delta_{eff}$ of section 2.5). Some of these curves have a bump: these could be because the structure described above enters their ball, and the radius for which this happens would then be the distance of the corresponding points to the structure (laminar distance of the section 3.3).

To further clarify this viewpoint, a trajectory of this system with $10^5$ points was generated and the dimension is computed for all points. The $10^5$ points were labelled as either laminar or chaotic, with the following steps:

1. restrict the curve for radii $R > 8$;

2. look for the maximum of this restricted curve;

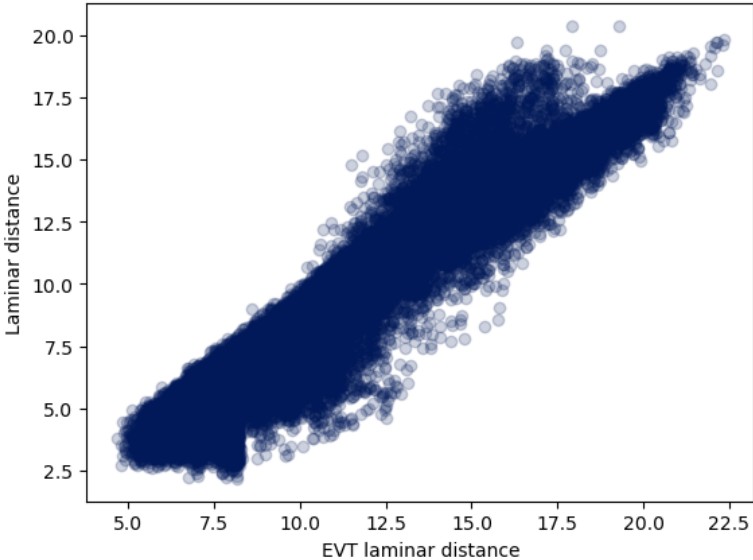

**Figure 14.** Scatter plots of the two ways to compute the distance to the laminar structure for chaotic points.

    3. if this maximum is not at the ends of this curve, and if the value of the dimension at the maximum is smaller than 12, the point is labelled as laminar.

Points not labelled as laminar are labelled as chaotic. Note that the laminar/chaotic points are not necessarily laminar/chaotic in the context of chaotic intermittency in dynamical systems, but we use this terminology to distinguish between points with different dimension characteristics.

    If the points we have labelled as laminar indeed form a geometric structure in phase space, the distance of chaotic points to this structure should correspond to a maximum of the dimension. Therefore, we proceed as in section 3.3: for each chaotic

point,

    1. we define the laminar distance as the minimum distance between the point and all the laminar points (in other words, using the set of points labelled as laminar as a representation of a laminar regime).

    2. we define the EVT laminar distance as the radius $R$ giving the maximum dimension.

    If our above labelling of laminar and chaotic points is meaningful, the two distances should agree. As shown in the scatter

plot of Fig. 14, there is a good agreement and we take this as a consistency check of our interpretation of the curves in the left panel of Fig. 13.

    Note that, for $32.8\%$ of the chaotic points, no distance could be computed using the second way (for the same reason as in section 3.3). Also, the fact that we find a structure in phase space, analog to the laminar structures of the previous sections, points to the fact that the Lorenz 96 system with $n = 50$ dimensions for $F = 4.9$ could be in an intermittent regime.

To summarize, we computed the dimension in this high-dimensional system and, as shown in the previous section 3.3, the dimension highly depends on the radius $R$ used to compute it, but this can be used to obtain some characterization of the geometry of the phase space: some points of the attractor are collected in a structure, much as the laminar points in section 3.3. The distance from the other points to this structure can be estimated.

## 4.2   RADCLIM dataset

We now present the results of the computation of the dimension for the RADCLIM dataset (see Appendix C for a description). Before computing the dimension, the images were upscaled to 14x14 for two reasons. The first is that the whole dataset is then obviously easier to work with. The second is based on the hope that the upscaled images would define a reduced and less complex attractor within a reduced phase space.

The upscaling to 14x14 images was done by averaging the neighbouring pixels, after the log-transform of the rain rate. This 565  is a way to take into account the multiplicative structure of the rain (Veneziano et al., 2006; Seed, 2003; Lovejoy and Schertzer, 2013) during the upscaling. As in Pulkkinen et al. (2019b), the zeros were transformed to -15 in logarithmic scale.

The geometry of the phase space defined by these images is quite particular. The value of each pixel is taken as an axis in phase space, which is therefore a 14x14 = 196 dimensional euclidean space. Since the minimum value for all pixels is -15, the phase space is actually restricted to the orthant defined by $x_i \geq -15 \ \forall i = 1, ..., 196$. The point $x_i = -15 \ \forall i$ corresponds to 570  images without any rain, which we call the dry event. There are 7150 such images (1.13% of all 630 008 images). Even among images with rain, most of them are close to the dry event. The histogram in Fig. 15 shows the distribution of all distances to the dry events (including the dry events themselves).

This histogram shows that the density of points in phase space decreases very quickly when getting away from the dry event. As a comparison, if the density of points $\rho$ was constant in phase space, the number of points whose distance to the dry event is 575  between $r$ and $r+dr$ would be $\approx \rho r^{n-1} dr = \rho r^{195} dr$. This means that the heights of the sticks in this histogram would grow as $r^{195}$, which is radically different from what is observed! The relationship between distances and volumes in high-dimensional spaces can be quite different from our 3-dimensional intuition.

For 2000 computation points, the dimension was computed for 40 different values of $R$. As before, 40 percentages were fixed (exponentially spaced between $8 \times 10^{-4}\% \rightarrow 5$ points and $10\% \rightarrow 63\,000$ points), and the radii $R$ were chosen to correspond 580  to the percentiles among all distances. Figure 16 shows the computed dimension as a function of the radius $R$ for 100 points: the left plot has a logarithmic scale for $R$ and the right plot has a linear scale for $R$, allowing to see more clearly the plot for small $R$ values or for large $R$ values. The color represents the 10-logarithm of the averaged convective rain rate (see below).

For some of the 2000 points, there is a bump in the dimension, which would point to the existence of some structure. We tried to apply the same procedure as in section 3.3 to compute the distance to the laminar regime. The procedure allowed to 585  get a laminar distance for only 487 out of 2000 points (for the same reason as in section 3.3: we discarded the estimation if the maximum dimension is on one of the ends of the range). It turns out that the laminar distance computed in this way is highly correlated to the distance to the dry event: see Fig. 17. As a consequence, the points labelled as laminar would simply be the

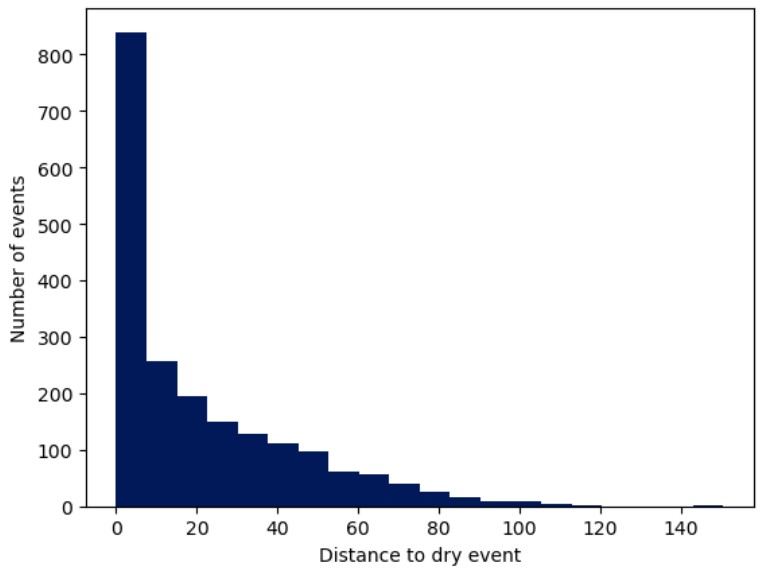

**Figure 15.** Histogram of the distances to the dry event for the RADCLIM dataset.

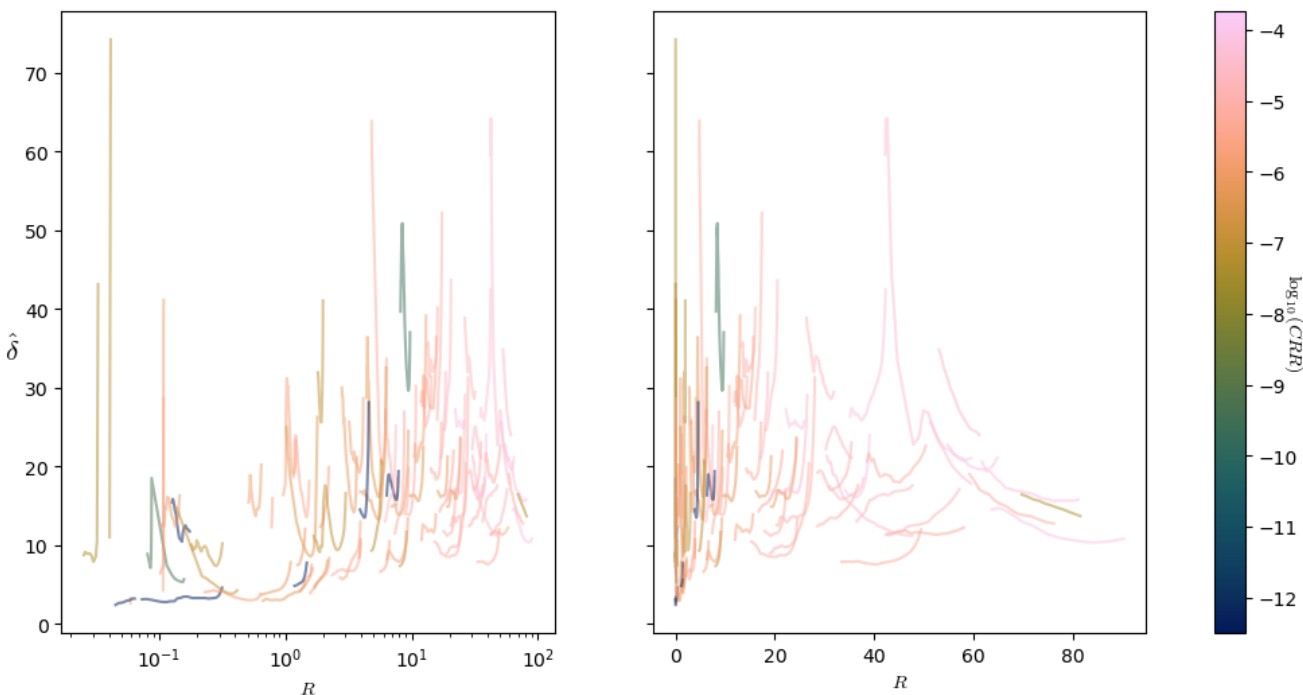

**Figure 16.** Plot of the dimension against the radius for 100 points for the RADCLIM data (NRMSE$< 0.4$ and $s > 5$). The color is the $10-$logarithm of the convective rain rate (CRR).

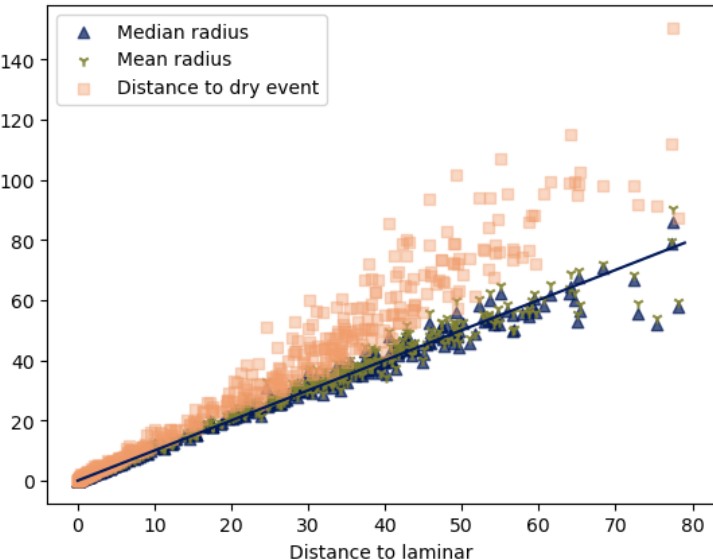

**Figure 17.** Scatter plot of the laminar distance on $x-$axis, with the median radius, the mean radius and the distance to the dry event on the $y-$axis.

dry events. The laminar distance as defined in this way does not really seem to contain any additional valuable information than the distance to dry events.

In order to check if there is any valuable information in the set of values of $R$ for any computation point, we looked for a way to aggregate the 40 values of $R$ we have for each computation point, and considered the mean and the median of $R$ for all computation points. Note that these quantities obviously depend on the way the percentages to compute the dimension for were chosen. Because of that, one could think that these mean and median would not be very informative. These values are however relatively stable: we checked for example that they do not change much if we use the 10 smallest radii that for each

point, instead of the 40 we have. This is because the extent of the range of values of radii we have for one computation point is small with respect to the actual values of the radii. This is also why the mean and the median do not differ much, and why they give a measure of the relevant values of the radius for each point. Figure 17 shows that there is a correlation of these two quantities (mean and median radii) with the laminar distance and the dry distance. This shows that these 4 quantities (median radius, mean radius, laminar distance, dry distance) are essentially the same.

We see here the difficulty to work with high-dimensional systems, as discussed in the section 2.4. The particular geometry of this phase space worsens even more the situation, because most of the points are collected near the dry event, as shown in the histogram of Fig. 15. Because of this, the dimension for all points can only be computed reliably on quite small ranges of values of $R$. For example, the points for which the curves are on the left of Fig. 16 have often $R$ approximatively within $[30, 40]$. This is a too small range of values of $R$ to properly interpret the computed values of $\delta$. The same happens for the

points for which the curves are on the right of the figure: they have often $R \in [80, 100]$.

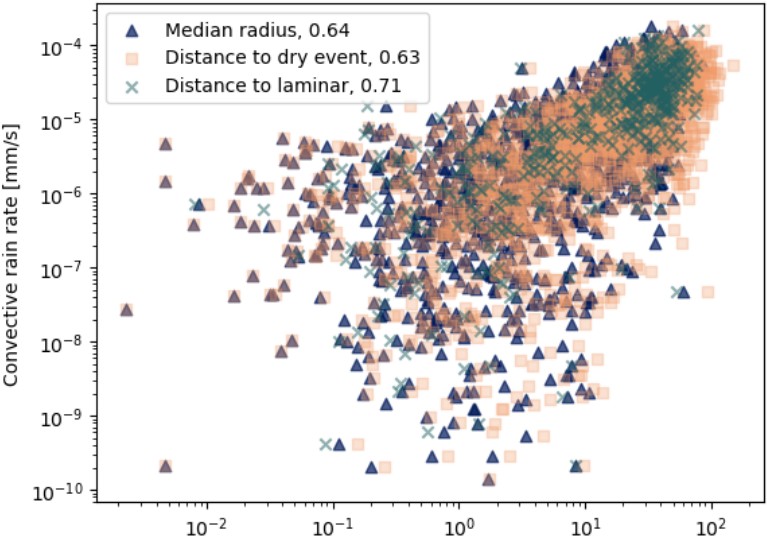

**Figure 18.** Scatter plots of the mean radius, the dry distance and the laminar distance ($x-$axis) with respect to mean convective rain rate (CRR) ($y-$axis). The numbers in the legend are the correlation coefficients of the 10-logarithm of each of these distances with $\log_{10}(CRR)$.

We used the convective available potential energy (CAPE), the convective rain rate (CRR) and the convective precipitation (CP) data from the ERA5 reanalysis to compare with the quantities computed in the phase space. For the region covered by the RADCLIM dataset, they come as 26x41 images with a 1-hour resolution. We computed the mean of these images in order to have one value for each hour. For each of the 2000 computation points, we associated the closest in time available value of the CAPE, the CP and the CRR. As suggested by the color grading in Fig. 16, one can find a correlation between the mean radius, the dry distance or the laminar distance on one hand, and the CRR on the other hand: Fig. 18 shows the corresponding scatter plots for the 2000 images.

Results with the CP instead of the CRR are very similar, while the correlations with the CAPE also exist but are not as good. The correlation of the CRR is the highest with the laminar distance but this is because of the restriction to some points. The laminar distance could indeed be computed for only 487 points out of 2000, while the mean radius and the dry distance can be computed for all the points. If we restrict the computation of the correlation between the CRR and the mean radius to the points for which a laminar distance was computed, one gets a correlation of $0.70$. The same happens for the correlation between the CRR and the dry distance.

We checked also if some information can be extracted from the mean value of the dimension: for each image, we computed the mean of the dimensions resulting from fits whose NRMSE score was below 0.4 and $s$ above 5. Figure 19 shows the histogram of the repartition of this mean dimension for our 2000 computation points (in orange) and the scatter plot of this mean dimension with the CRR (in color grading from dark blue to orange). A complementary 2D histogram of the mean dimension with to the CRR is shown in Appendix C (Fig. C1).

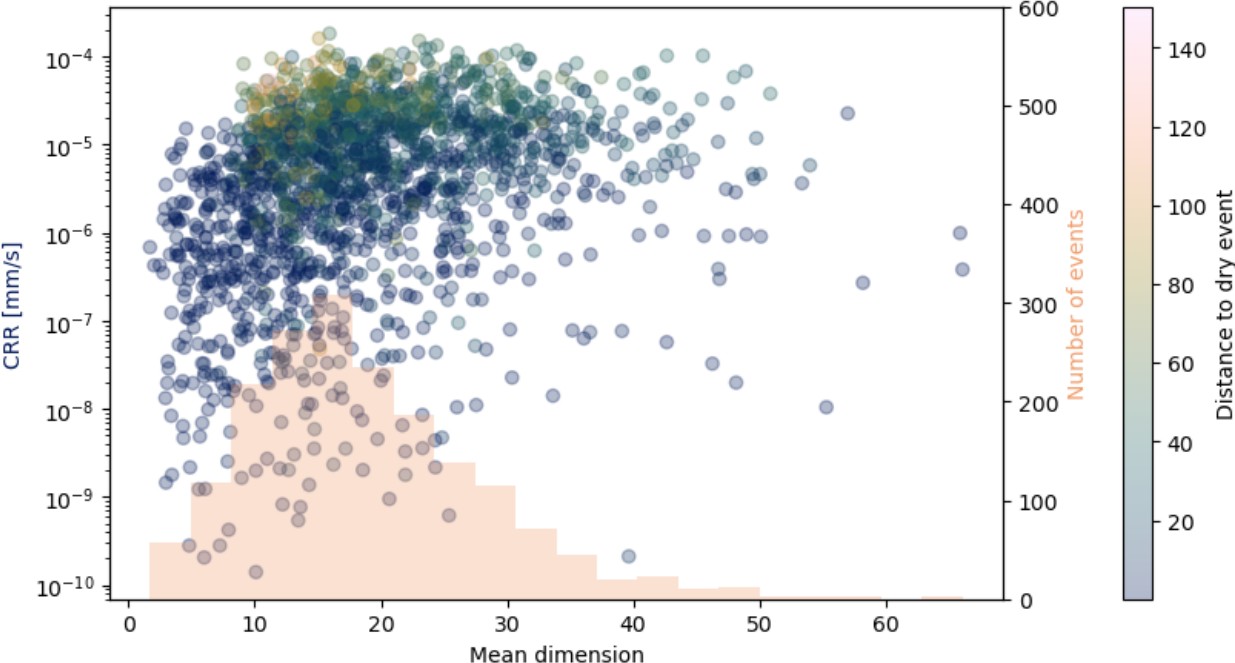

**Figure 19.** Histogram of the repartition of the mean dimension for our 2000 computation points (in orange) and scatter plot of this mean dimension with the CRR (in color scale). The color is the distance to the dry event.

A few things can be noted here:

– The histogram has a nice peak around a mean dimension of 15-20.

    – The relationship with the CRR is not very clear, except that there is generally less variability in the CRR when the dimension is higher.

    – As expected, images the farther away from the dry event have the higher CRR, but those images do not have the highest values of the mean dimension: their mean dimension is rather between 10 and 20.

– When the mean dimension is below 10, the distance to the dry event is always relatively small. Vice versa, for big enough distances, the dimension is always above 10. This can also be seen from Fig. 16 taking into account that the distance to the dry event is almost the same as $R$.

To summarize, this analysis of the results of the computed dimension for the RADCLIM dataset shows that it is possible to compute reliably the dimension for some points and for some radii $R$, but the results are quite difficult to interpret. This 635   is essentially because of the limited number of data with respect to the number of dimensions at play. The estimated local dimensions ranges between 10 and 30, with a peak around 15-20. Not surprisingly, we observed a link between the convective rain rate and the distance to the dry event.

## 5 Summary and discussion

As emphasized in the introduction, results on the local dimension of attractors of dynamical systems predict that almost all points have the same dimension in the asymptotic limit of infinite number of points and infinitely small radius $R$. Because these limits are impossible to reach, we studied the behavior of the local dimension for finite $R$ and showed that it allows to detect geometrical structures in phase space of chaotic dynamical systems. The main visible feature of this detection, is that the dimension has a peak for the value of $R$ corresponding to the entrance of a geometric structure inside $B(R)$.

When working with such tools, one always faces the question to know if the estimation of the dimension is reliable or not. This question is linked to that of the maximal dimension that one can estimate with a given number of points inside $B(R)$. We systematically used the NRMSE score and the quantity $s$ to tackle this problem. The development leading to the definition of $s$ brought some light on the problem of the maximal measurable dimension using EVT techniques. This gives some falsifiability methods, which were lacking before, as was noticed in Datseris et al. (2023).

In short, the dimension is estimated as the exponent of the power law $C(r) \sim r^\delta$ over a some range $[0, R]$. If the NRMSE score is good enough, a $10\%$ accuracy for $\delta$ relatively to $\hat{\delta}$ is achieved with $N \gtrsim 400$ points. However, a high exponent in the scaling $C(r) \sim r^\delta$ (i.e. a high dimension) will not be visible over a long interval $[0, R]$ if there are not enough points. This is why high dimensions may be estimated, but only on limited ranges. There is no contradiction with the argument of Eckmann and Ruelle (1992) : for a fixed range $[0, R]$, the number of points needed to measure a dimension indeed grow exponentially with the dimension. The same applies for the correlation dimension: for a fixed number of points, higher dimensions can be measured if the range $[0, R]$ decreases.

As shown, the dimension depends on the finite radius $R$ of the ball $B(R)$ used to compute the dimension. This implies that one cannot simply choose some small percentage (as $2\%$ or $5\%$), compute $R$ as the corresponding percentile and get a unique value for the dimension. In fact, different finite values of $R$ may lead to very different values for the dimension.

To understand what is captured by the computed dimension at a given scale $R$, one has to compare $R$ with the other local geometrical scales on the attractor. In this work, these scales were mainly set by some geometrical structures, but Perinelli et al. (2023) showed that the curvature could also set some scale, and the idea that the scale of the noise could play a role was raised by Little et al. (2017). We also identified that estimations of the dimension can be affected when the density of points is not constant. This often leads to an overestimation of the dimension. The value of the dimension for a given $R$ can be affected by yet other phenomena, and it is important to recognize which ones are at play to interpret correctly the dimension.

For the RADCLIM dataset, one difficulty is that the number of points is rather limited so that, for each computation point, values of the dimension could only be computed in a very limited range of values of $R$. This was expected from the analysis of section 2.4, and this makes difficult the interpretation of the value of the dimension for this high-dimensional system. Some interesting conclusions could however be gathered from the analysis, in particular that the range of dimension is between 10 and 30 with a peak at 15-20.

There exists other ways to compute the local dimension, such as the Lyapunov dimension (Kaplan and Yorke (1979), see Ott (2002) for a textbook review) and the dimension induced by the delay coordinate method (delay embedding dimension, Packard

et al. (1980), see Abarbanel (1996) for a textbook review). The dimensions estimated using the correlation dimension, which we argued is conceptually equivalent to the EVT dimension, and using the EVT dimension itself are compared the Lyapunov dimension and to the delay embedding dimension in Datseris et al. (2023). Note also that, for these two latter definitions of the dimension, there is no equivalent to the scale $R$ of the EVT dimension and the correlation dimension. Because of that, we do not expect future comparison study (if any) to be able to recover the interpretation of the $R$-dependence of the dimension proposed here for the Lyapunov dimension and the delay embedding dimension.

## Appendix A: Lorenz 63 system

The Lorenz 63 system is defined by the following equations (Lorenz, 1963)

$$
\begin{cases}
\dot{x} &= \sigma(y - x) \\
\dot{y} &= \rho x - y - xz \\
\dot{z} &= xy - \beta z
\end{cases}
\tag{A1}
$$

where $\sigma, \beta$ and $\rho$ are constant. Usual values are $\sigma = 10$, $\beta = 8/3$ and $\rho = 28$. In this configuration, the system is known to be chaotic.

The same system for $\rho = 166.5$ was also considered ($\sigma$ and $\beta$ being unchanged). This system is known to be intermittent, meaning that it follows regular and almost periodic patterns for some periods of time (so-called "laminar" phases), alternating with other periods where it seems to behave randomly (the "chaotic bursts"), see Ott (2002); Schuster and Just (2006); Elaskar and Rio (2017) about intermittent dynamical systems. A trajectory is displayed in Fig. A1.

As was first noted in Pomeau and Manneville (1980), this system undergoes a bifurcation at $\rho \approx 166.07$: the attractor is first periodic but disappears through a saddle-node bifurcation (Sparrow, 1982).

## Appendix B: Lorenz 96 system

The Lorenz 96 system (Lorenz, 1996) is a dynamical system with $n$ variables $x_i$ for $i = 1, \ldots, n$ ($n \geq 4$). The evolution equations for the $x_i$'s are

$$
\dot{x}_i = (x_{i+1} - x_{i-2})x_{i-1} - x_i + F
\tag{B1}
$$

where index $i$ is understood as periodic: $x_{-1} = x_{n-1}$, $x_0 = x_n$ and $x_{n+1} = x_1$. The parameter $F$ is a forcing constant.

This system for $n = 4$ goes through a saddle-node bifurcation at $F \approx 11.83$ (Sterk and van Kekem, 2017; van Kekem and Sterk, 2018) and is intermittent for slightly higher values of $F$ (see Appendix A for a brief introduction on intermittency). Figure B1 shows that the estimated value of $\delta$ is maximum when the laminar distance is equal to $R$, as in Fig. 11.

We also considered the Lorenz 96 system with $n = 12$ for $F = 4.4$. We found a bifurcation at $F \approx 4.25$ with an intermittent behavior after this value. The laminar regime for this system forms a higher dimensional structure in phase space and the trajectory in laminar phases is not almost periodic. One cannot proceed as we did for Lorenz 63 with $\rho = 166.5$ to get a

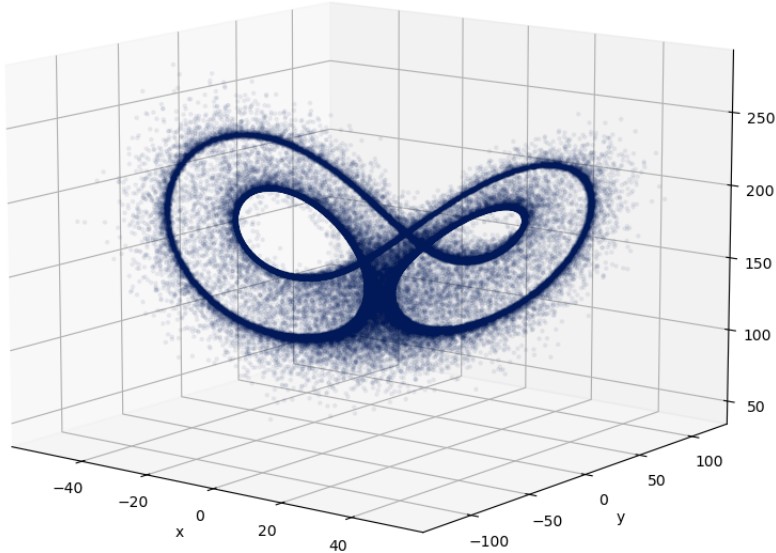

**Figure A1.** Lorenz 63 attractor for $\rho = 166.5$. The remaining of the attractor before the bifurcation is visible.

representation of the laminar regime. Instead, we used a trajectory for $F = 4.2$ to represent the laminar regime. The problem with this approach is that the laminar regime after the bifurcation has moved and expanded in phase space with respect to the attractor at $F = 4.2$. The peak of the dimension when the laminar distance is equal to $R$ is still clearly visible in Fig. B2.

In complement to the analysis for the Lorenz 96 system with $n = 50$ dimensions in section 4, we computed the local dimension for the same system for $F = 6$. We used a trajectory of $10^7$ points. The left panel of Fig. B3 displays the dimension as a function of $R$ for 50 computation points (restricted to NRMSE< 0.4 and $s > 6$), while the right panel is the scatter plot of the NRMSE vs $s$ for all fits. The behavior of the dimension against $R$ is the same for all points and we conclude that our method suggests that there is no salient geometric structure in phase space for that parameter value.

## Appendix C: RADCLIM dataset

The RADCLIM radar dataset (Goudenhoofdt and Delobbe, 2016; Journée et al., 2023) is a high horizontal and temporal resolution quantitative precipitation estimation in Belgium and its surroundings. It is based on radar measurements, which are merged with rain gauges measures.

The time resolution is 5 minutes and we used 6 years of product. A few images are missing in the dataset, which has in total 630 008 radar images. The images are 700x700, with each pixel representing a square of 1kmx1km.

As a complement to Fig. 19, Fig. C1 is a 2D histogram of the mean dimension against the CRR. 5000 computation points were used (instead of 2000 in section 4.2) in order to have a more reliable histogram.

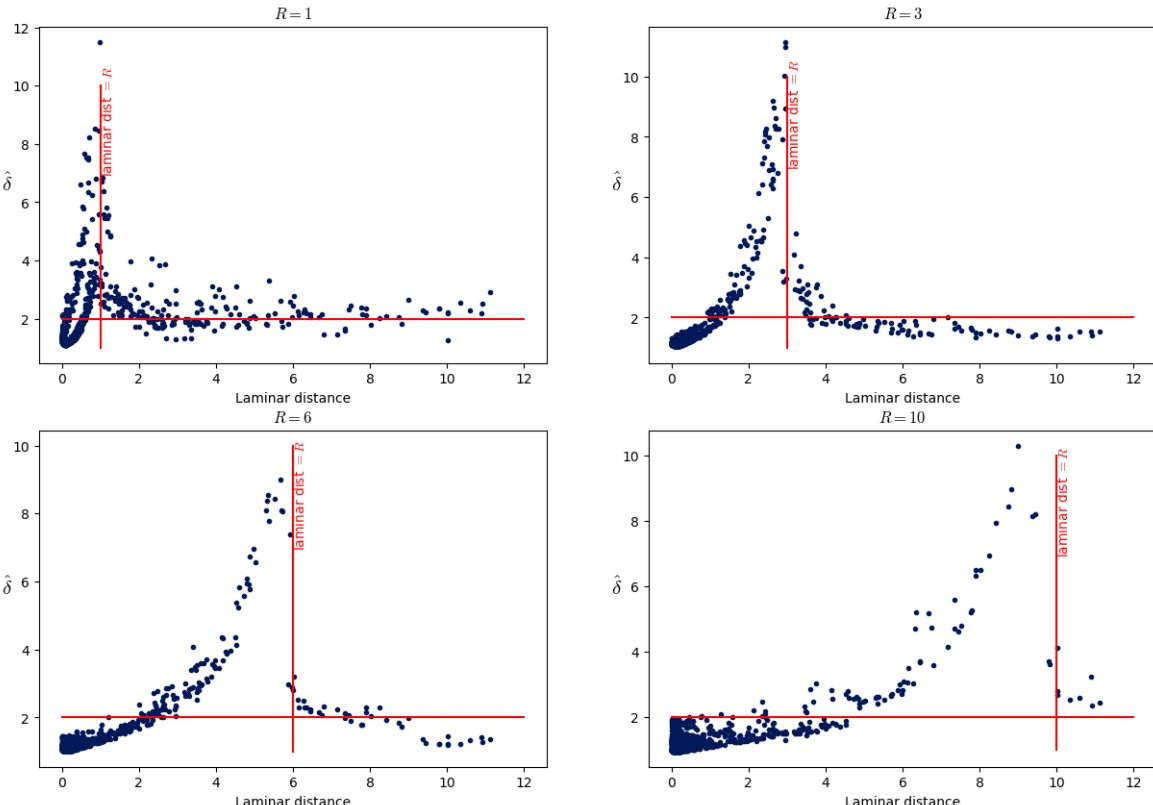

**Figure B1.** Same as Fig. 11 but for the Lorenz 96 system with $n = 4$ and $F = 11.87$ ($10^6$ points in the trajectory). The vertical red lines mark the value of the laminar distance which is equal to the radius $R$ and the horizontal line marks the values 2 for $\hat{\delta}$.

*Author contributions.* MB and SV designed the work done. MB made the computations and wrote the text. SV helped with discussions about the results and the proofreading of the text.

*Competing interests.* Stéphane Vannitsem is a member of the editorial board of Nonlinear Processes in Geophysics.

*Acknowledgements.* The authors are grateful to Davide Faranda and Tommaso Alberti for discussions about the interpretation of the local dimension. This work is supported by the Belgian Federal Science Policy under contract B2/233/P2/PRECIP-PREDICT.

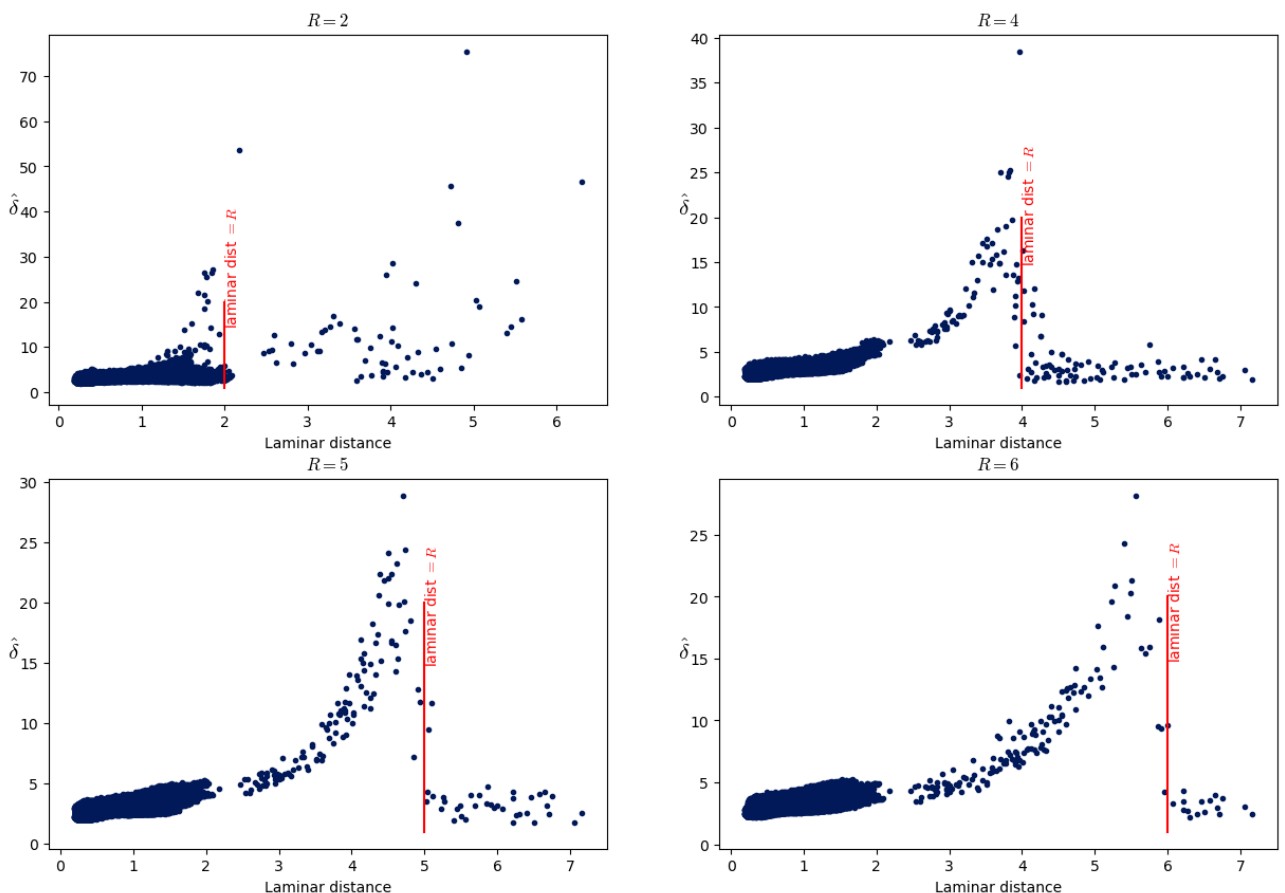

**Figure B2.** Same as Fig. 11 but for the Lorenz 96 system with $n = 12$ and $F = 4.4$ ($10^5$ points in the trajectory). The vertical red lines mark the value of the laminar distance which is equal to the radius $R$.

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

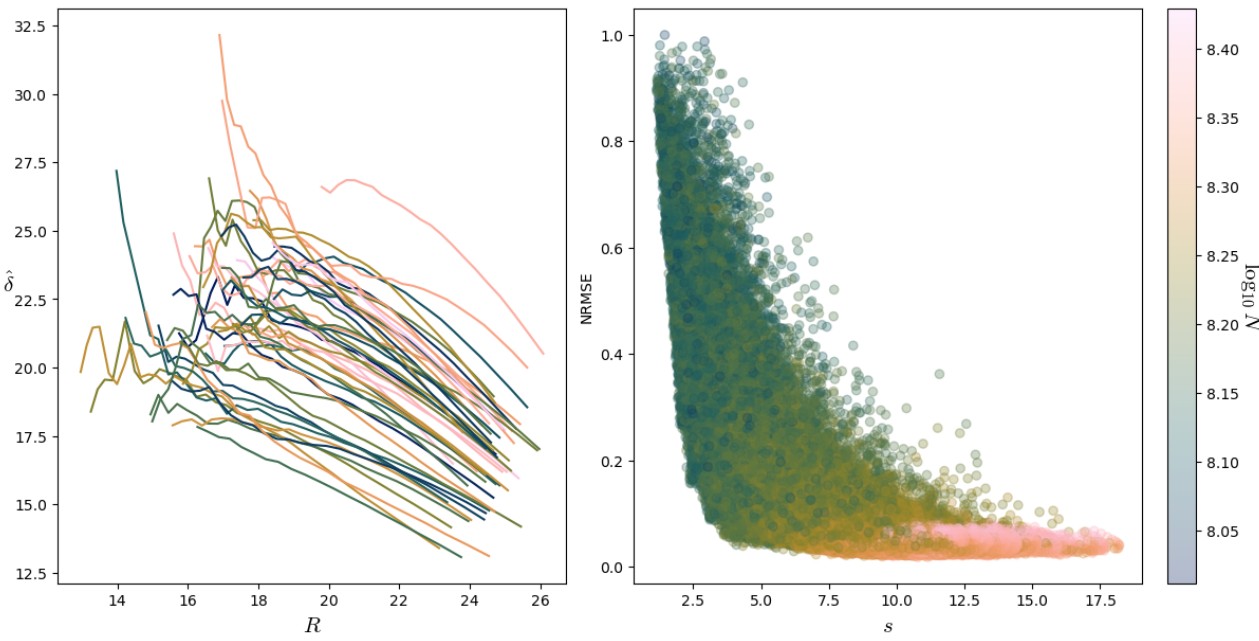

**Figure B3.** For the Lorenz 96 system with $n = 50$ dimensions, same as Fig. 13 but for $F = 6$. Left: dimension $\hat{\delta}$ as a function of $R$ for 50 computation points (NRMSE$< 0.4$ and $s > 6$). Right: scatter plot of the NRMSE vs $s$ (all fits).

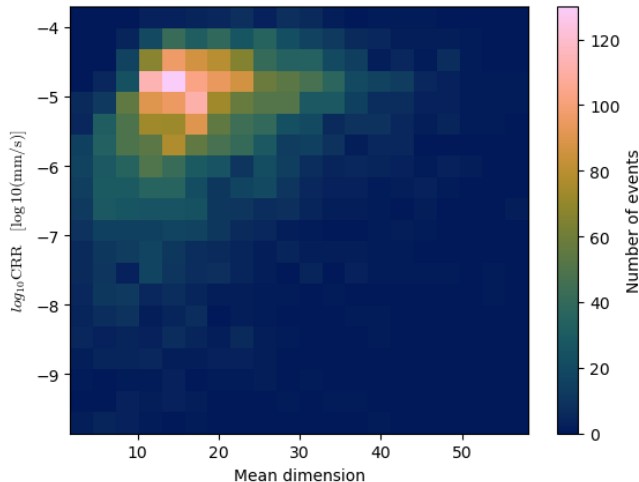

**Figure C1.** 2D histogram of the repartition of the mean dimension for 5000 computation points with the associated CRR.

Bowler, N., Pierce, C., and Seed, A.: STEPS: A probabilistic precipitation forecasting scheme which merges an extrapolation nowcast with downscaled NWP, Quarterly Journal of the Royal Meteorological Society, 132, 2127 – 2155, https://doi.org/10.1256/qj.04.100, 2007.

Caby, T.: Extreme value theory for dynamical systems, with applications in climate and neuroscience, Theses, Université de Toulon ; Università degli studi dell' Insubria (Come, Italie). Facolta' scienze matematiche, fisiche e naturali, https://theses.hal.science/tel-02473235, 2019.

Camastra, F. and Staiano, A.: Intrinsic dimension estimation: Advances and open problems, Information Sciences, 328, 26–41, https://doi.org/https://doi.org/10.1016/j.ins.2015.08.029, 2016.

Datseris, G., Kottlarz, I., Braun, A. P., and Parlitz, U.: Estimating fractal dimensions: A comparative review and open source implementations, Chaos: An Interdisciplinary Journal of Nonlinear Science, 33, https://doi.org/10.1063/5.0160394, 2023.

De Luca, P., Messori, G., Pons, F. M. E., and Faranda, D.: Dynamical systems theory sheds new light on compound climate extremes in Europe and Eastern North America, Quarterly Journal of the Royal Meteorological Society, 146, 1636–1650, https://doi.org/https://doi.org/10.1002/qj.3757, 2020.

Eckmann, J.-P. and Ruelle, D.: Fundamental limitations for estimating dimensions and Lyapunov exponents in dynamical systems, Physica D: Nonlinear Phenomena, 56, 185–187, https://doi.org/https://doi.org/10.1016/0167-2789(92)90023-G, 1992.

Elaskar, S. and Rio, E.: New Advances on Chaotic Intermittency and its Applications, ISBN 978-3-319-47837-1, https://doi.org/10.1007/978-3-319-47837-1, 2017.

Erba, V., Gherardi, M., and Rotondo, P.: Intrinsic dimension estimation for locally undersampled data, Scientific Reports, 9, https://doi.org/10.1038/s41598-019-53549-9, 2019.

Falk, M., Hüsler, J., and Reiss, R.: Laws of Small Numbers: Extremes and Rare Events, Springer Basel, ISBN 9783034800099, 2010.

Faranda, D., Lucarini, V., Turchetti, G., and Vaienti, S.: Extreme value theory for singular measures, Chaos: An Interdisciplinary Journal of Nonlinear Science, 22, 023 135, https://doi.org/10.1063/1.4718935, 2012.

Faranda, D., Messori, G., and Yiou, P.: Dynamical proxies of North Atlantic predictability and extremes, Scientific Reports, https://doi.org/10.1038/srep41278, 2017.

Faranda, D., Messori, G., and Vannitsem, S.: Attractor dimension of time-averaged climate observables: insights from a low-order ocean-atmosphere model, Tellus A: Dynamic Meteorology and Oceanography, https://doi.org/10.1080/16000870.2018.1554413, 2019.

Faranda, D., Bourdin, S., Ginesta, M., Krouma, M., Noyelle, R., Pons, F., Yiou, P., and Messori, G.: A climate-change attribution retrospective of some impactful weather extremes of 2021, Weather and Climate Dynamics, 3, 1311–1340, https://doi.org/10.5194/wcd-3-1311-2022, 2022.

Faranda, D., Pascale, S., and Bulut, B.: Persistent anticyclonic conditions and climate change exacerbated the exceptional 2022 European-Mediterranean drought, Environmental Research Letters, 18, 034 030, https://doi.org/10.1088/1748-9326/acbc37, 2023.

Foresti, L., Puigdomènech Treserras, B., Nerini, D., Atencia, A., Gabella, M., Sideris, I. V., Germann, U., and Zawadzki, I.: A quest for precipitation attractors in weather radar archives, Nonlinear Processes in Geophysics, 31, 259–286, https://doi.org/10.5194/npg-31-259-2024, 2024.

Germann, U. and Zawadzki, I.: Scale Dependence of the Predictability of Precipitation from Continental Radar Images. Part II: Probability Forecasts, Journal of Applied Meteorology, 43, 74 – 89, https://doi.org/10.1175/1520-0450(2004)043<0074:SDOTPO>2.0.CO;2, 2004.

Germann, U., Zawadzki, I., and Turner, B.: Predictability of Precipitation from Continental Radar Images. Part IV: Limits to Prediction, Journal of the Atmospheric Sciences, 63, 2092 – 2108, https://doi.org/10.1175/JAS3735.1, 2006.

Golay, J. and Kanevski, M. F.: A new estimator of intrinsic dimension based on the multipoint Morisita index, Pattern Recognit., 48, 4070–4081, https://api.semanticscholar.org/CorpusID:31779757, 2014.

Goudenhoofdt, E. and Delobbe, L.: Generation and Verification of Rainfall Estimates from 10-Yr Volumetric Weather Radar Measurements, Journal of Hydrometeorology, 17, 1223 – 1242, https://doi.org/10.1175/JHM-D-15-0166.1, 2016.

Grassberger, P. and Procaccia, I.: Measuring the strangeness of strange attractors, Physica D: Nonlinear Phenomena, 9, 189–208, https://doi.org/https://doi.org/10.1016/0167-2789(83)90298-1, 1983.

James, F. E.: Statistical Methods in Experimental Physics; 2nd ed., World Scientific, Singapore, 2006.

Journée, M., Goudenhoofdt, E., Vannitsem, S., and Delobbe, L.: Quantitative rainfall analysis of the 2021 mid-July flood event in Belgium, Hydrology and Earth System Sciences, 27, 3169–3189, https://doi.org/10.5194/hess-27-3169-2023, 2023.

Kantz, H. and Schreiber, T.: Nonlinear Time Series Analysis, Cambridge University Press, 2 edn., 2003.

Kaplan, J. L. and Yorke, J. A.: Chaotic behavior of multidimensional difference equations, in: Functional differential equations and approximations of fixed points. Lecture notes in mathematics Vol. 730, edited by Peitgen, H. O. and Walter, H. O., Springer, Berlin, 1979.

Little, A. V., Maggioni, M., and Rosasco, L.: Multiscale geometric methods for data sets I: Multiscale SVD, noise and curvature, Applied and Computational Harmonic Analysis, 43, 504–567, https://doi.org/https://doi.org/10.1016/j.acha.2015.09.009, 2017.

Lorenz, E. N.: Deterministic Nonperiodic Flow., Journal of the Atmospheric Sciences, 20, 130–148, https://doi.org/10.1175/1520-0469(1963)020<0130:DNF>2.0.CO;2, 1963.

Lorenz, E. N.: Predictability: A problem partly solved, in: Proc. Seminar on predictability, vol. 1, Reading, 1996.

Lovejoy, S. and Schertzer, D.: The Weather and Climate: Emergent Laws and Multifractal Cascades, Cambridge University Press, 2013.

Lucarini, V., Faranda, D., Moreira Freitas, A. C., Freitas, J. M., Mark, H., Tobias, K., Nicol, M., Todd, M., and Vaienti, S.: Extremes and Recurrence in Dynamical Systems, Pure and Applied Mathematics: A Wiley Series of Texts, Monographs and Tracts, Wiley Interscience, https://hal.science/hal-02886423, 2016.

Ott, E.: Chaos in Dynamical Systems, Cambridge University Press, 2 edn., 2002.

Packard, N. H., Crutchfield, J. P., Farmer, J. D., and Shaw, R. S.: Geometry from a Time Series, Phys. Rev. Lett., 45, 712–716, https://doi.org/10.1103/PhysRevLett.45.712, 1980.

Perinelli, A., Iuppa, R., and Ricci, L.: Estimating the correlation dimension of a fractal on a sphere, Chaos, Solitons & Fractals, 173, 113 632, https://doi.org/https://doi.org/10.1016/j.chaos.2023.113632, 2023.

Pesin, Y.: Dimension Theory in Dynamical Systems: Contemporary Views and Applications, Chicago Lectures in Mathematics, University
of Chicago Press, ISBN 9780226662213, 1997.

Pierce, C., Seed, A., Ballard, S., Simonin, D., and Li, Z.: Nowcasting, in: Doppler Radar Observations, edited by Bech, J. and Chau, J. L., chap. 4, IntechOpen, Rijeka, https://doi.org/10.5772/39054, 2012.

Pomeau, Y. and Manneville, P.: Intermittent transition to turbulence in dissipative dynamical systems, Communications in Mathematical Physics, 74, https://doi.org/10.1007/BF01197757, 1980.

Pons, F. M. E., Messori, G., Alvarez-Castro, M. C., and Faranda, D.: Sampling hyperspheres via extreme value theory: implications for measuring attractor dimensions, Journal of Statistical Physics, https://doi.org/10.1007/s10955-020-02573-5, 2020.

Pons, F. M. E., Messori, G., and Faranda, D.: Statistical performance of local attractor dimension estimators in non-Axiom A dynamical systems, Chaos: An Interdisciplinary Journal of Nonlinear Science, 33, 073 143, https://doi.org/10.1063/5.0152370, 2023.

Pulkkinen, S., Chandrasekar, V., and Harri, A.-M.: Stochastic Spectral Method for Radar-Based Probabilistic Precipitation Nowcasting,
Journal of Atmospheric and Oceanic Technology, 36, 971 – 985, https://doi.org/10.1175/JTECH-D-18-0242.1, 2019a.

Pulkkinen, S., Nerini, D., Pérez Hortal, A. A., Velasco-Forero, C., Seed, A., Germann, U., and Foresti, L.: Pysteps: an open-source Python library for probabilistic precipitation nowcasting (v1.0), Geoscientific Model Development, 12, 4185–4219, https://doi.org/10.5194/gmd-12-4185-2019, 2019b.

Russell, D. A., Hanson, J. D., and Ott, E.: Dimension of Strange Attractors, Phys. Rev. Lett., 45, 1175–1178, https://doi.org/10.1103/PhysRevLett.45.1175, 1980.

Schuster, H. and Just, W.: Deterministic Chaos: An Introduction, Wiley, ISBN 9783527606412, 2006.

Scott, D.: Multivariate Density Estimation: Theory, Practice, and Visualization, Wiley Series in Probability and Statistics, Wiley, ISBN 9780471697558, 2015.

Seed, A. W.: A Dynamic and Spatial Scaling Approach to Advection Forecasting, Journal of Applied Meteorology, 42, 381 – 388, https://doi.org/10.1175/1520-0450(2003)042<0381:ADASSA>2.0.CO;2, 2003.

Sparrow, C.: The Lorenz Equations: Bifurcations, Chaos, and Strange Attractors, Applied Mathematical Sciences, Springer New York, ISBN 9781461257677, 1982.

Sterk, A. and van Kekem, D.: Predictability of Extreme Waves in the Lorenz-96 Model Near Intermittency and Quasi-Periodicity, Complexity, 2017, 1–14, https://doi.org/10.1155/2017/9419024, 2017.

Takens, F.: On the numerical determination of the dimension of an attractor, in: Dynamical Systems and Bifurcations, edited by Braaksma, B. L. J., Broer, H. W., and Takens, F., pp. 99–106, Springer Berlin Heidelberg, Berlin, Heidelberg, ISBN 978-3-540-39411-2, 1985.

Theiler, J.: Spurious dimension from correlation algorithms applied to limited time-series data, Phys. Rev. A, 34, 2427–2432, https://doi.org/10.1103/PhysRevA.34.2427, 1986.

Theiler, J.: Statistical precision of dimension estimators, Phys. Rev. A, 41, 3038–3051, https://doi.org/10.1103/PhysRevA.41.3038, 1990.

van Kekem, D. L. and Sterk, A. E.: Travelling waves and their bifurcations in the Lorenz-96 model, Physica D: Nonlinear Phenomena, 367, 38–60, https://doi.org/https://doi.org/10.1016/j.physd.2017.11.008, 2018.

Veneziano, D., Langousis, A., and Furcolo, P.: Multifractality and rainfall extremes: A review, Water Resources Research, 42, https://doi.org/https://doi.org/10.1029/2005WR004716, 2006.

Zawadzki, I., Morneau, J., and Laprise, R.: Predictability of Precipitation Patterns: An Operational Approach, Journal of Applied Meteorology and Climatology, 33, 1562 – 1571, https://doi.org/10.1175/1520-0450(1994)033<1562:POPPAO>2.0.CO;2, 1994.