# Peer review of "Finite-size local dimension as a tool for extracting geometrical properties of attractors of dynamical systems"

_EGUsphere, 2024_

## Author Response (AR1)

**Authors' response**

**Manuscript egusphere-2024-3915**
**"Finite-size local dimension as a tool for extracting geometrical properties of attractors of dynamical systems"**
**M. Bonte and S. Vannitsem**

**To the editor**

Dear editor,

Please find below the reviewer's comments and our replies. We took into account all their comments and modified the manuscript accordingly. We also rewrote some parts of the manuscript.

The changes are highlighted in the 'Author's track-changes file', except for new Figures or Figures that have been reproduced. We reproduced Fig. A1 and added the results of the computation of the local dimension for the Lorenz 96 system for $n = 50$ dimensions and $F = 6$ in Fig. B3. Figures 1 and 2 have also been modified.

Thank you for handling our manuscript.

Best regards,

Martin Bonte
For the authors

**Reply to Referee 1:**

In high-dimensional chaotic dynamics, unstable dimensions typically vary along a trajectory. In dynamical system literature, such phenomena are studied in various ways, including hetero-dimensional cycles, unstable dimension variability, hetero-chaos, riddling bifurcation, and blowout bifurcation. The paper focuses on the finite-size local dimension computed using the Extreme Value Theory (EVT). The local dimension or a similar object has been investigated in various ways, as listed above, for over 20-30 years, although the use of EVT is relatively new. The paper first details the 63-Lorenz and the 96-Lorenz systems, and two-parameter cases are discussed for the former, Then, the technique is applied to the RADCLIM dataset to obtain an estimate of the dimensions. The obtained results and the methods used in the paper are interesting.

*Thank you very much for your constructive comment. Please find below our replies (italicized) to your comments.*

Comments:

The local dimension can be estimated using several ways, including the finite-time Lyapunov exponents and the Kaplan-Yorke formula or the delay coordinate of an observable variable. The authors should compare the methods and clarify the advantages/disadvantages of the EVT method.

*This comparison has already been done by Datseris et al. (2023). We believe in addition that the paper is already quite long. As suggested by Reviewer 2, we then propose not to enter into a detailed discussion of other approaches to compute the dimension. Rather we suggest to explicitly mention at the end of the conclusion the comparison done by Datseris et el. (2023):*

*'There exists other ways to compute the  dimension, such as the Lyapunov dimension (Kaplan and Yorke, (1979), see Ott (2002) for a textbook review) and the dimension induced by the delay coordinate method (delay embedding dimension, Packard et al. (1980), see Abarbanel (1996) for a textbook review). The dimensions estimated using the correlation dimension, which we argued is conceptually equivalent to the EVT dimension, and the EVT dimension itself are compared the Lyapunov dimension and to the delay embedding dimension in Datseris et al. (2023). Note also that, for these two latter definitions of the dimension, there is no equivalent to the scale R of the EVT dimension and the correlation dimension. Because of that, we do not expect future comparison study (if any) to be able to recover the interpretation of the R-dependence of the dimension proposed here for the Lyapunov dimension and the delay embedding dimension.'*

The two parameters of the Lorenz 63 system are suitable, as the classical rho=28 is known to (singular) hyperbolic, and the other rho=166.5 has tangencies between stable and unstable manifolds, which break the hyperbolicity. However, the Lorenz 96 system with n=50 and F=4.9 is not suitable for an example of relatively high dimensional dynamics, as it rarely shows unstable dimension variability. I recommend reanalyzing F=6 or higher in detail with dimensions n greater than or equal to 8.  See Danforth and Yorke, Physical Review Letters 2006, which shows unstable dimension variability in the Lorenz 96 system.

*We extend the analysis done for F=4.9 in the Lorenz 96 model to the case F=6. We computed the local dimension for 50 points in a trajectory of $10^7$ points. A plot of the dimension as a function of R is attached. The behavior of the dimension against R is the same for all points and we conclude that our method suggests that there is no salient geometric structure for that parameter value. We propose to add this figure in a supplement to the paper.*

I recommend the authors invite those familiar with dynamical system theory for careful revision throughout the paper. There are many inappropriate expressions on dynamical system theory, which lead to misunderstanding for the readers. For example, the object in Fig. A1 does not seem to be an attractor that is an invariant set.

*Figure A1 is indeed a plot the attractor of the Lorenz 63 system with rho = 166.5 with too few points. Thank you for pointing out to this. We reproduced it but now with much more points. It is also attached to this reply.*

*Thank you for the typos you mentioned, we have proofread the paper and have hopefully caught all remaining typos.*

References:

Kaplan, J.L., Yorke, J.A. (1979). Chaotic behavior of multidimensional difference equations. In: Peitgen, HO., Walther, HO. (eds) Functional Differential Equations and Approximation of Fixed Points. Lecture Notes in Mathematics, vol 730. Springer, Berlin, Heidelberg. https://doi.org/10.1007/BFb0064319

N.H. Packard, J.P. Crutchfield, J.D. Farmer, R.S. Shaw, Geometry from a time series. Phys. Rev. Lett. 45(9), 712–716 (1980)

Abarbanel, H. D. I.: Analysis of Observed Chaotic Data, pp. 69–93, Springer New York, New York, NY, ISBN 978-1-4612-0763-4, https://doi.org/10.1007/978-1-4612-0763-4_5, 1996

Datseris, G., Kottlarz, I., Braun, A. P., and Parlitz, U.: Estimating fractal dimensions: A comparative review and open source implementations, 685 Chaos: An Interdisciplinary Journal of Nonlinear Science, 33, https://doi.org/10.1063/5.0160394, 2023

**Reply to referee 2**

The essence of classical Extreme Value Theory (EVT) lies in establishing the statistical properties of the extremes of observables generated by stochastic processes. This theory is critically important for a wide range of mathematical and physical applications, for reasons that are readily apparent. Over the past 25 years, EVT has been successfully applied, both theoretically and empirically, to the study of chaotic dynamical systems, demonstrating promising potential for systems that meet specific assumptions, such as strong mixing properties.

Notably, the local geometric properties of certain invariant sets in deterministic dynamical systems can be inferred through EVT, as the distribution of observables on these sets is ultimately governed by their geometric structure. This paper contributes to this line of research by examining the pointwise dimension. Specifically, it investigates the dimension by focusing on a local ball-shaped region and analyzing the statistical behavior as a function of the distance from the center of the ball. The paper presents intriguing new results that merit publication.

However, the manuscript contains numerous misleading expressions and ambiguous explanations, which create unnecessary challenges for readers who are not thoroughly familiar with dynamical systems and EVT. Furthermore, there are several grammatical errors and typographical mistakes, many of which are noticeable even to non-native speakers like myself. I agree with the first referee that substantial revisions are required, particularly in the phrasing and mathematical language.

*Thank you very much for your input that considerably helped shaping much better the manuscript. We tried to clarify better the mathematical concept used in the manuscript, together with the language in the perspective of potential non-specialized readers. Please find below our replies (italicized) to your comments.*

Below, I outline some of the inappropriate expressions and typographical errors I have identified:

4. **Line 43:** "percentage of points" → This lacks context. What do you mean by the "percentage of points"?

*We propose to replace 'In Datseris et al. (2023), the discussion about the percentage of points needed suggests that N (the number of points at a distance smaller than R from the computation point ζ) just needs to be higher than 100-1000.'*
*By*

*'Datseris et al. (2023) suggests that N (the number of points at a distance smaller than R from the computation point ζ) just needs to be higher than 100-1000.'*

6 and 7. **Lines 94-95:** "In the limit of infinite trajectories" → Do you mean infinitely long trajectories, assuming ergodicity on the attractor? A concise and precise definition would be helpful.

"this measure can be approximated as the number C(R) of points inside B(R), divided by the total number of points in the trajectory" → This sentence is vague. My interpretation is that, for any given trajectory, after some initial "burn-in" time allowing the trajectory to converge to the attractor,

the invariant measure can be approximated as the number C(R) of iterations within B(R), divided by the total number of iterations in the trajectory.

*We indeed mean infinitely long trajectories, assuming ergodicity for the natural measure $\mu$. Your interpretation about the second part of the sentence is what we meant. We propose to replace the paragraph*

*'In the limit of infinite trajectories, this measure can be approximated as the number C(R) of points inside B(R), divided by the total number of points in the trajectory. In other words, the pointwise dimension Dp($\zeta$) can thus be interpreted as a characterization of the growth rate of the number of points that one should find inside a ball B(R) centered around $\zeta$ (for infinitely small R).'*

*by the three following paragraphs:*

*'The measure $\mu$ is often chosen to be the natural measure of the system: given any long enough trajectory originating from a typical initial condition of the system, $\mu(A)$ is defined for any subset A of the phase space as the fraction of points of the trajectory inside A, or similarly as the fraction of time spend by the system in A (see Ott (2002) and Kantz and Schreiber (2004)). This means that the whole set of points of the trajectory is as if it had been sampled from the measure $\mu$. Some points may have to be discarded at the beginning of the trajectory in order to ensure that the solution has converged on the attractor.*

*This natural measure is invariant by definition: $\mu(A) = \mu(\Phi_t^{-1}(A))$, where $\Phi_t$ is the flow of the system. This is because each point in the trajectory has one antecedent point, so that the number of points in A is the same than in $\Phi_t^{-1}(A)$. It is also ergodic if the attractor cannot be decomposed in two distinct invariant sets. In practice, one can also assume ergodicity by assuming that the system is always on the same invariant subset (for example because we observe one and only one trajectory, as in the case of climate). As stated in the introduction, the dimension $D_p(\zeta)$ is constant for almost all points when the measure is ergodic (Pesin, 1997, Ott, 2002, Pons et al., 2020).*

*The above discussion implies that, given a long enough trajectory, $\mu(B(R))$ can be approximated as the number C(R) of points inside B(R), divided by the total number of points in the trajectory. It follows that the pointwise dimension $D_p$(zeta) can be interpreted as a characterization (see (5) below for a precise statement) of the increase of the number points C(R) that one should find inside a ball B(R) centered around $\zeta$ (for infinitely small R).'*

8. **Line 96:** "the pointwise dimension Dp($\zeta$) can thus be interpreted as a characterization of the growth rate of the number of points that one should find inside a ball B(R) centered around $\zeta$ (for infinitely small R)" → This is true when the trajectory length is sufficiently large and held constant.

*More precisely, under the assumptions detailed in this paragraph, C(R) is approximated by eq. (5). We added a reference to eq. (5) in the last of the three paragraphs proposed in the reply to the points 6 and 7.*

9. **Line 98:** "if they form a smooth Dp($\zeta$)-dimensional surface" → A set of points cannot form a D-dimensional hypersurface. You likely mean they all lie on the hypersurface.

*Indeed, we will change "if they form a smooth $D_p(\zeta)$-dimensional surface" for "if they span a smooth $D_p(\zeta)$-dimensional surface"*

10. **Line 115:** "The distances measured in phase space do not precisely match with r or R: if the surface is curved" → Do you mean that R is the radius length measured along the geodesics on the attractor, provided that the attractor is well-approximated locally by a differentiable manifold?

*This is indeed what we mean by 'along the surface of the attractor' at line 116. For more clarity, we propose to replace 'if the surface is curved, the points on the boundary of B(R) will have a distance along the surface of the attractor bigger than R (see Fig. 1)'*

*By*

*'for eq. (5) to hold, R has to be measured on the manifold of the attractor, but this is not possible when our representation of the attractor is a set of points. Instead, we measure distances in phase space and, if the attractor is curved, there can be a mismatch with the distances measured along the manifold of the attractor (see Fig. 1).'*

12. **Line 119:** "smaller than some typical scale of the curvature" → More precisely, do you mean when curvature ×R is sufficiently small?

*We will add 'given for example by the inverse of the curvature itself' after 'smaller than some typical scale characterizing the curvature'.*

13. **Line 144, eq. (10):** Check your math. From Eq.(5), it seems $C'(r)dr \sim r^{\delta-1}dr/(\delta-1)!$, not $\delta r^{\delta-1}dr$. The final expression in Eq.(11) remains unaffected, but accuracy is always preferable.

*The important part of this formula is the r-dependence of c(r) and C'(r) and we meant that we were only keeping track of this through the use of the sign $\sim$. To be precise, we will replace eq. (10) by $c(r)dr \sim C'(r)dr = C^{\delta}(0)r^{\delta-1} dr/(\delta-1)!$.*

14. **Line 146, eq. (11):** Although mathematically correct, a physical interpretation of c(r) would be helpful: if a point on the trajectory lies within B(R), c(r)dr is the probability of finding it in the shell between radii r and r+dr.

*An equivalent interpretation is stated just above eq. (10) (lines 142-143). For clarity, we will add below eq. (11) the following sentence 'With such a normalization for c(r), it can be interpreted as follows: if we select a point randomly inside B(R), c(r)dr is the probability that its distance to $\zeta$ is between r and r + dr (for r < R).'*

15. **Line 147**: "The usual EVT framework for the dimension" → Instead of assuming familiarity with the EVT approach in Lucarini et al. (2012), it would be helpful to explain the rationale here: a point on a trajectory is considered "extreme" if it lies within B(R) for sufficiently small R. The "threshold" in the POT framework corresponds to R, and "peak over threshold" means radii smaller than R. See Eq.(9) in Lucarini et al. (2012).

*We propose to replace 'The usual EVT framework for the dimension is found by using one of the following transformations:'*

*By*

*'The usual EVT framework for the dimension is formulated within the POT framework. The observable whose extreme value distribution is studied in this context is often one of the following function of r:'*

*After 'The parameter K can be freely chosen, and α and γ have to be positive.',*

*We propose also to add*

*'Given a point ζ in phase space, the points corresponding to extremes are those whose distances to ζ are smaller than R. The threshold of the POT approach is given by $T\_a = g\_a(R)$ (a = 1, 2, 3), where $g\_a$ is the function that was chosen from the above three. Note that EVT usually defines extremes as* high *values of an observable while, in terms of the distance r to ζ, extremes are defined as* small *values of r.'*

*Thank you for the typos you mentioned, we have proofread the paper and have hopefully caught all remaining typos.*

References:

Ott, E.: Chaos in Dynamical Systems, Cambridge University Press, 2 edn., 2002

Pesin, Y.: Dimension Theory in Dynamical Systems: Contemporary Views and Applications, Chicago Lectures in Mathematics, University of Chicago Press, ISBN 9780226662213, 1997.

Pons, F. M. E., Messori, G., Alvarez-Castro, M. C., and Faranda, D.: Sampling hyperspheres via extreme value theory: implications for measuring attractor dimensions, Journal of Statistical Physics, https://doi.org/10.1007/s10955-020-02573-5, 2020.